# Tensor Normal Training for Deep Learning Models

**Yi Ren, Donald Goldfarb**
Department of Industrial Engineering and Operations Research
Columbia University
New York, NY 10027
{yr2322, goldfarb}@columbia.edu

## Abstract

Despite the predominant use of first-order methods for training deep learning models, second-order methods, and in particular, natural gradient methods, remain of interest because of their potential for accelerating training through the use of curvature information. Several methods with non-diagonal preconditioning matrices, including KFAC [34], Shampoo [18], and K-BFGS [15], have been proposed and shown to be effective. Based on the so-called *tensor normal* (TN) distribution [31], we propose and analyze a brand new approximate natural gradient method, *Tensor Normal Training* (TNT), which like Shampoo, only requires knowledge of the shape of the training parameters. By approximating the probabilistically based Fisher matrix, as opposed to the empirical Fisher matrix, our method uses the block-wise covariance of the sampling based gradient as the pre-conditioning matrix. Moreover, the assumption that the sampling-based (tensor) gradient follows a TN distribution, ensures that its covariance has a Kronecker separable structure, which leads to a tractable approximation to the Fisher matrix. Consequently, TNT's memory requirements and per-iteration computational costs are only slightly higher than those for first-order methods. In our experiments, TNT exhibited superior optimization performance to state-of-the-art first-order methods, and comparable optimization performance to the state-of-the-art second-order methods KFAC and Shampoo. Moreover, TNT demonstrated its ability to generalize as well as first-order methods, while using fewer epochs.

## 1   Introduction

First-order methods are currently by far the most popular and successful optimization methods for training deep learning models. Stochastic gradient descent (SGD) [39] uses the (stochastic) gradient direction to guide its update at every iteration. Adaptive learning rate methods, including AdaGrad [11], RMSprop [20], and Adam [23], scale each element of the gradient direction (possibly modified to incorporate momentum) by the square root of the second moment of each element of the gradient. These first-order methods use little curvature information to "pre-condition" the gradient direction; SGD uses an identity pre-conditioning matrix, whereas the others use a diagonal matrix.

On the other hand, second-order methods attempt to greatly accelerate the optimization process by exploring the rich curvature information of the problem. Traditional second-order methods such as Newton's method, BFGS [6, 13, 14, 41], and limited-memory BFGS (L-BFGS) [28], without modification, are not practical in a deep learning setting, because these methods require enormous amounts of memory and computational effort per iteration due to the huge number of parameters such models have. Some second-order methods have been proposed to deal with the non-convexity and stochasiticity of objective functions arising in machine learning (see e.g. [36, 7, 16, 44]), but directly using these methods to train deep learning models still requires large amounts of memory and computing resources.

35th Conference on Neural Information Processing Systems (NeurIPS 2021).

Recently, there has been considerable advancement in the development of second-order methods that are suitable for deep learning models with huge numbers of parameters. These methods usually approach pre-conditioning of the gradient in a modular way, resulting in block-diagonal pre-conditioning matrices, where each block corresponds to a layer or a set of trainable parameters in the model. Inspired by the idea of the natural gradient (NG) method [1], [34] proposed KFAC, an NG method that uses a Kronecker-factored approximation to the Fisher matrix as its pre-conditioning matrix that can be applied to multilayer perceptrons, and which has subsequently been extended to other architectures, such as convolutional neural networks [17] and recurrent neural networks [35]. Kronecker-factored preconditioners [15, 38] based on the structure of the Hessian and quasi-Newton methods have also been developed. Despite the great success of these efficient and effective second-order methods, developing such methods requires careful examination of the structure of the preconditioning matrix to design appropriate approximations for each type of layer in a model.

Another well-recognized second-order method, Shampoo [18, 3], extends the adaptive learning rate method AdaGrad, so that the gradient is pre-conditioned along every dimension of the underlying tensor of parameters in the model, essentially replacing the diagonal pre-conditioning matrix of the adaptive learning rate methods by a block diagonal Kronecker-factored matrix which can be viewed as an approximation to a fractional power of the empirical Fisher (EF) matrix. However, while estimating the Fisher matrix, in a deep learning setting, by the EF matrix saves some computational effort, it usually does not capture as much valuable curvature information as the Fisher matrix [26].

Variants of the normal distribution, i.e. the matrix-normal distribution [9] and the tensor-normal distribution [31], have been proposed to estimate the covariance of matrix and higher-order tensor observations, respectively. By imposing a Kronecker structure on the covariance matrix, the resulting covariance estimate requires a vastly reduced amount of memory, while still capturing the interactions between the various dimensions of the respective matrix or tensor. Iterative MLE methods for estimating the parameters of matrix-normal and tensor-normal distributions have been examined in e.g. [12, 31], and various ways to identify the unique representation of the distribution parameters have been proposed in [43, 10]. However, to the best of our knowledge, this advanced statistical methodology has not been used to develop optimization methods for deep learning. In this paper, we describe a first attempt to do this and demonstrate its great potential.

**Our Contributions.** In this paper, we propose a brand new approximate natural gradient (NG) method, Tensor-Normal Training (TNT), that makes use of the tensor normal distribution to approximate the Fisher matrix. Significantly, the TNT method can be applied to any model whose training parameters are a collection of tensors without knowing the exact structure of the model.

To achieve this, we first propose a new way, that is suitable for optimization, to identify the covariance parameters of tensor normal (TN) distributions, in which the average eigenvalues of the covariance matrices corresponding to each of the tensor dimensions are required to be the same (see Section 3).

By using the Kronecker product structure of the TN covariance, TNT only introduces mild memory and per-iteration computational overhead compared with first-order methods. Also, TNT's memory usage is the same as Shampoo's and no greater than KFAC's, while its per-iteration computational needs are no greater than Shampoo's and KFAC's (see Section 5).

The effectiveness of TNT is demonstrated on deep learning models. Specifically, on standard autoencoder problems, when optimization performance is compared, TNT converges faster than the benchmark first-order methods and roughly the same rate as the benchmark second-order methods. Moreover, on standard CNN models, when generalization is concerned, TNT is able to achieve roughly the same level of validation accuracy as the first-order methods, but using far fewer epochs (see Section 6).

We also prove that, if the statistics used in TNT can be estimated ideally, it converges to a stationary point under mild assumptions (see Section 4).

## 2   Preliminaries

**Supervised Learning.** Throughout this paper, we consider the classic supervised learning setting where we learn the parameters $\theta$ of a model, by minimizing $\mathcal{L}(\theta) = \frac{1}{N} \sum_{i=1}^{N} l(y_i, f_\theta(x_i))$, where $\{(x_i, y_i)\}_{i=1}^{N}$ denotes a given dataset ($x_i$ being the input to the model and $y_i$ being the target), $f_\theta(x_i)$ denotes the output of the model when $x_i$ is provided as the input, and $l$ denotes a loss function

(e.g. least-squares loss for regression and cross entropy loss for classification) that measures the discrepancy between the model output $f_\theta(x_i)$ and the target $y_i$.

**Natural Gradient Method and the Fisher Matrix.** In a first-order method, say SGD, the updating direction is always derived from an estimate to the gradient direction $\nabla_\theta \mathcal{L}(\theta)$. In a natural gradient (NG) method [1], however, the Fisher matrix is used as a pre-conditioning matrix that is applied to the gradient direction. As shown in [34], the Fisher matrix is defined as

$$F = \mathbb{E}_{x \sim Q_x, y \sim p(\cdot|x,\theta)} \left[ \nabla_\theta \log p(y|x,\theta) \left( \nabla_\theta \log p(y|x,\theta) \right)^\top \right], \tag{1}$$

where $Q_x$ is the data distribution of $x$ and $p(\cdot|x,\theta)$ is the density function of the conditional distribution defined by the model with a given input $x$.

In many cases, such as when $p$ is associated with a Gaussian distribution and the loss function $l$ measures least-squares loss, or when $p$ is associated with a multinomial distribution and $l$ is cross-entropy loss, $\log p$ is equivalent to $l$ (see e.g. [33, 34]). Hence, if $\mathcal{D}\theta$ denotes the gradient of $l$ w.r.t. $\theta$ for a given $x$ and $y$, we have that $F = \mathbb{E}_{x \sim Q_x, y \sim p}[\mathcal{D}\theta \mathcal{D}\theta^\top]$. Consequently, one can sample $x$ from $Q_x$ and perform a forward pass of the model, then sample $y$ from $p(\cdot|x,\theta)$, and perform a backward pass to compute $\mathcal{D}\theta$, and then use $\mathcal{D}\theta \mathcal{D}\theta^\top$ to estimate $F$. We call $\mathcal{D}\theta$ a *sampling-based gradient*, as opposed to the *empirical gradient* $\nabla_\theta l(y_i, f_\theta(x_i))$ where $(x_i, y_i)$ is one instance from the dataset.

It is worth noting that the first moment of $\mathcal{D}\theta$ is zero. To see this, note that, with given $x$,

$$\mathbb{E}_{y \sim p}[\nabla_\theta \log p(y|x,\theta)] = \int \nabla_\theta \log p(y|x,\theta) p(y|x,\theta) dy = \int \nabla_\theta p(y|x,\theta) dy$$

$$= \nabla_\theta \left( \int p(y|x,\theta) dy \right) = \nabla_\theta 1 = 0.$$

Hence, $\mathbb{E}_{x \sim Q_x, y \sim p}[\mathcal{D}\theta] = \mathbb{E}_{x \sim Q_x} \{ \mathbb{E}_{y \sim p}[\nabla_\theta \log p(y|x,\theta)] \mid x \} = 0$. Thus, the Fisher matrix $F$ can be viewed as the covariance matrix of $\mathcal{D}\theta$. Note that the empirical Fisher CANNOT be viewed as the covariance of the empirical gradient, because the first moment of the latter is, in general, NOT zero.

**Tensor-Normal Distribution.** The development of our new method makes use the so-called *tensor-normal* distribution [31, 10]:

**Definition 1.** *An arbitrary tensor $G \in \mathbb{R}^{d_1 \times \cdots \times d_k}$ is said to follow a tensor normal (TN) distribution with mean parameter $M \in \mathbb{R}^{d_1 \times \cdots \times d_k}$ and covariance parameters $U_1 \in \mathbb{R}^{d_1 \times d_1}$, ..., $U_k \in \mathbb{R}^{d_k \times d_k}$ if and only if $\overline{vec}(G) \sim Normal(\overline{vec}(M), U_1 \otimes \cdots \otimes U_k)$.*

In the above definition, the $\overline{vec}$ operation refers to the *vectorization* of a tensor, whose formal definition can be found in Sec A in the Appendix. Note that *matrix-normal* distribution can be viewed as a special case of TN distribution, where $k = 2$. Compared with a regular normal distribution, whose covariance matrix has $\prod_{i=1}^{k} d_i^2$ elements, the covariance of a $k$-way tensor-normal distribution is stored in $k$ smaller matrices with a total number of elements equal to $\sum_{i=1}^{k} d_i^2$.

To estimate the covariance submatrices $U_1, \ldots, U_k$, the following property (e.g., see [10]) is used:

$$\mathbb{E}[G^{(i)}] = U_i \cdot \prod_{j \neq i} \text{tr}(U_j), \tag{2}$$

where $G^{(i)} := \text{mat}_i(G)\text{mat}_i(G)^\top \in \mathbb{R}^{d_i \times d_i}$ denotes the *contraction* of $G$ with itself along all but the $i$th dimension and $\text{mat}_i$ refers to *matricization* of a tensor (see Section A for the formal definitions). By (2), we can sample $G$ to obtain estimates of the $G^{(i)}$'s, and hence, estimates of the $U_i$'s. The complexity of computing $G^{(i)}$ is $d_i \prod_{j=1}^{k} d_j$, which is also far less than the complexity of computing $\overline{vec}(G)\overline{vec}(G)^\top$ needed to estimate the covariance of a regular normal distribution.

## 3 Tensor-Normal Training

In this section, we propose Tensor-Normal Training (TNT), a brand new variant of the natural gradient (NG) method that makes use of the tensor-normal distribution.

## 3.1 Block Diagonal Approximation

In this paper, we consider the case where the parameters of the model $\theta$ consists of multiple tensor variables $W_1, ..., W_L$, i.e. $\theta = \left(\overline{\text{vec}}(W_1)^\top, ..., \overline{\text{vec}}(W_L)^\top\right)^\top$. This setting is applicable to most common models in deep learning such as multi-layer perceptrons, convolutional neural networks, recurrent neural networks, etc. In these models, the trainable parameter $W_l$ ($l = 1, \ldots, L$) come from the weights or biases of a layer, whether it be a feed-forward, convolutional, recurrent, or batch normalization layer, etc. Note that the index $l$ of $W_l$ refers to the index of a tensor variable, as opposed to a layer.

To obtain a practical NG method, we assume, as in KFAC and Shampoo, that the pre-conditioning Fisher matrix is block diagonal. To be more specific, we assume that each block corresponds to the covariance of a tensor variable in the model. Hence, the approximate Fisher matrix is:

$$F \approx \text{diag}_{l=1}^L \left\{ \mathbb{E}_{x \sim Q_x, y \sim p} \left[ \overline{\text{vec}}(\mathcal{D}W_l)(\overline{\text{vec}}(\mathcal{D}W_l))^\top \right] \right\} = \text{diag}_{l=1}^L \left\{ \text{Var}(\overline{\text{vec}}(\mathcal{D}W_l)) \right\}.$$

The remaining question is how should one approximate $\text{Var}(\overline{\text{vec}}(\mathcal{D}W_l))$ for $l = 1, ..., L$.

## 3.2 Computing the Approximate Natural Gradient Direction by TNT

We consider a tensor variable $W \in \mathbb{R}^{d_1 \times \cdots \times d_k}$ in the model and assume that $G := \mathcal{D}W \in \mathbb{R}^{d_1 \times \cdots \times d_k}$ follows a TN distribution with zero mean and covariance parameters $U_1, ..., U_k$ where $U_i \in \mathbb{R}^{d_i \times d_i}$. Thus, the Fisher matrix corresponding to $W$ is $F_W = \mathbb{E}_{x \sim Q_x, y \sim p}[\text{Var}(\overline{\text{vec}}(G))] = U_1 \otimes \cdots \otimes U_k$. Loosely speaking, the idea of relating the Fisher matrix to the covariance matrix of some normal distribution has some connections to Bayesian learning methods and interpretations of NG methods (see e.g., [22]). Let $\nabla_W \mathcal{L} \in \mathbb{R}^{d_1 \times \cdots \times d_k}$ denote the gradient of $\mathcal{L}$ w.r.t. $W$. The approximate NG updating direction for $W$ is computed as

$$F_W^{-1} \overline{\text{vec}}(\nabla_W \mathcal{L}) = (U_1^{-1} \otimes \cdots \otimes U_k^{-1}) \overline{\text{vec}}(\nabla_W \mathcal{L}) = \overline{\text{vec}}\left(\nabla_W \mathcal{L} \times_1 U_1^{-1} \times_2 \cdots \times_k U_k^{-1}\right), \quad (3)$$

where $\times_i$ ($i = 1, ..., k$) denotes a mode-$i$ product (see Section A in the Appendix). Note that the last equality of (3) makes use of the following proposition, which also appears in [18] (see Sec A in the Appendix for a proof):

**Proposition 1.** *Let $G \in \mathbb{R}^{d_1 \times \cdots \times d_k}$ and $U_i \in \mathbb{R}^{d_i \times d_i}$ for $i = 1, ..., k$. Then, we have*

$$\left(\otimes_{i=1}^k U_i\right) \overline{vec}(G) = \overline{vec}(G \times_1 U_1 \times_2 U_2 \cdots \times_k U_k). \quad (4)$$

To summarize, the generic Tensor-Normal Training algorithm is:

---
**Algorithm 1** Generic Tensor-Normal Training (TNT)

---
**Require:** Given batch size $m$, and learning rate $\alpha$
1: **for** $t = 1, 2, \ldots$ **do**
2:    Sample mini-batch $M_t$ of size $m$
3:    Perform a forward-backward pass over $M_t$ to compute the mini-batch gradient
4:    Perform another backward pass over $M_t$ with $y$ sampled from the predictive distribution to compute $G_l = \mathcal{D}W_l$ ($l = 1, ..., L$) averaged across $M_t$
5:    **for** $l = 1, ...L$ **do**
6:        Estimate $\mathbb{E}[G_l^{(i)}]$ ($i = 1, ..., k_l$) from $G_l$
7:        Determine $U_1^{(l)}, ..., U_{k_l}^{(l)}$ from $\mathbb{E}[G_l^{(1)}], ..., \mathbb{E}[G_l^{(k_l)}]$
8:        Compute the inverses of $U_1^{(l)}, ..., U_{k_l}^{(l)}$
9:        Compute the updating direction $p_l$ by (3)
10:       $W_l = W_l - \alpha \cdot p_l$.
11:   **end for**
12: **end for**

---

## 3.3 Identifying the Covariance Parameters of the Tensor Normal Distribution

By (2), $U_i$ can be inferred from $\mathbb{E}[G^{(i)}]$ up to a constant multiplier. However, different sets of multipliers can generate the same $F$, i.e. the same distribution. This is less of a problem if one

only cares about $F$. However, we need $F^{-1}$ to compute the approximate natural gradient. That is, we first must choose a representation of $F = c(\tilde{U}_1 \otimes \cdots \otimes \tilde{U}_k)$ (see below), and then compute $F^{-1} = c^{-1}((\tilde{U}_1 + \epsilon I)^{-1} \otimes \cdots \otimes (\tilde{U}_k + \epsilon I)^{-1})$ with a proper choice of $\epsilon > 0$, where $\epsilon I$ plays a damping role in the preconditioning matrix. In this case, different representations of $F$ will lead to different $F^{-1}$.

The statistics community has proposed various representations for $\tilde{U}_i$'s. For example, [43] imposed that $c = 1$ and the first element of $\tilde{U}_i$ to be one for $i = 1, ..., k - 1$, whereas [10] imposed that $\text{tr}(\tilde{U}_i) = 1$ for $i = 1, ..., k$. Although these representations have nice statistical properties, they are not ideal from the perspective of inverting the covariance for use in a NG method in optimization.

We now describe one way to determine $\tilde{U}_1, ..., \tilde{U}_k$, and $c$ from $\mathbb{E}[G^{(1)}], ..., \mathbb{E}[G^{(k)}]$. In particular, we first set $c = 1$, so that $F^{-1}$ has a constant upper bound $\epsilon^{-k} I$. We then require that $\frac{\text{tr}(\tilde{U}_i)}{d_i}$ is constant w.r.t $i$. In other words, the average of the eigenvalues of each of the $\tilde{U}_i$'s is the same. This helps the $\tilde{U}_i$'s have similar overall "magnitude" so that a suitable $\epsilon$ can be found that works for all dimensions. Moreover, this shares some similarity with how KFAC splits the overall damping term between KFAC matrices, although KFAC adjusts the damping values, whereas TNT adjusts the matrices. A bit of algebra gives the formula

$$\tilde{U}_i = \frac{\mathbb{E}[G^{(i)}]}{c_0^{k-1} \prod_{j \neq i} d_j}, \tag{5}$$

where $c_0 = \left( \frac{\text{tr}(\mathbb{E}[G^{(i)}])}{\prod_j d_j} \right)^{1/k}$.

### 3.4   Comparison with Shampoo and KFAC

Shampoo, proposed in [18], and later modified and extended in [3], is closely related to TNT. Both methods use a block-diagonal Kronecker-factored preconditioner based on second-order statistics of the gradient and are able to handle all sorts of tensors, and hence, can be applied to all sorts of deep neural network models, easily and seamlessly. The major differences between them are:

(i) The TN distribution cannot be directly applied to EF, which is used in Shampoo, because the empirical gradient does not have a zero mean; hence its covariance and second moment are different. It is also believed that EF does not capture as much valuable curvature information as Fisher [26].
(ii) Using the statistics $\mathbb{E}[G^{(i)}]$'s, TNT approximates the Fisher matrix as the covariance of the block-wise sampling-based gradients assuming that they are TN distributed. On the other hand, Shampoo computes $1/2k$-th power of the statistics of each direction of the tensor-structured empirical gradient and forms a preconditioning matrix from the Kronecker product of them. It is unclear to us how to interpret statistically such a matrix other than by its connection to EF. We further note that Shampoo was developed as a Kronecker-factored approximation to the full-matrix version of AdaGrad [11], whereas TNT was developed as a NG method using a TN-distributed approximation to the Fisher matrix.
(iii) TNT computes the updating direction using the inverse (i.e. power of $-1$) of the Kronecker factors of the approximate Fisher matrix, whereas Shampoo uses the $-1/2k$-th power[1] of the Kronecker factors of the EF matrix.

Another method related to TNT is KFAC [34, 17], which, like TNT, uses Fisher as its preconditioning matrix. Their major differences are:

(i) KFAC develops its approximation based on the structure of the gradient and Fisher matrix for each type of layer. Admittedly, this could lead to better approximations. But it is relatively hard to implement (e.g. one need to store some intermediate variables to construct the KFAC matrices). Also, if new types of layers with different structures are considered, one needs to develop suitable Kronecker factorizations, i.e., KFAC matrices. On the contrary, TNT, like Shampoo, is a model-agnostic method, in the sense that, TNT can be directly applied as long as the shape of the tensor variables are specified.

---

[1]In [3], for autoencoder problems involving tensors of order 2, the power was set to $-\frac{\alpha}{2}$, where $\alpha \in [0, 1]$ was treated as a hyper-parameter which required tuning, and was set to $\alpha = 1$ after tuning.

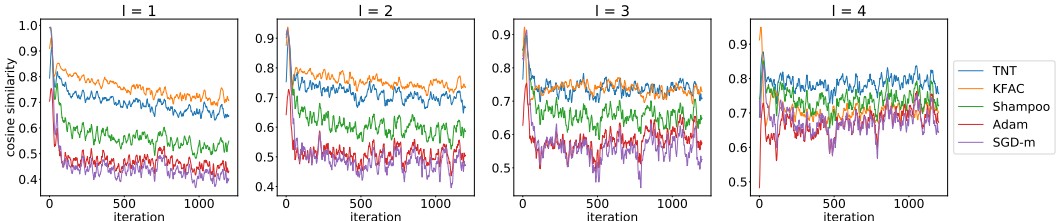

Figure 1: Cosine similarity between the directions produced by the methods shown in the legend and that of a block Fisher method. The algorithms were run on a $16 \times 16$ down-scaled MNIST [27] dataset and a small feed-forward NN with layer widths 256-20-20-20-20-20-10 described in [34]. As in [34], we only show the middle four layers.

(ii) Each block of TNT corresponds to a tensor variable whose shape needs to be specified, whereas each block of KFAC corresponds to all variables in a layer. For example, for a linear or convolutional layer, the KFAC block would correspond to the Fisher of both its weights and bias (and their correlation), whereas TNT would produce two blocks corresponding to the weights and bias, respectively.

In order to gain more insight into how well TNT approximates the Fisher matrix compared with other methods, we computed the cosine similarity between the direction produced by each method and that by a block Fisher method, where each block corresponded to one layer's full Fisher matrix in the model (see Figure 1). For all methods shown in Figure 1, we always followed the trajectory produced by the block Fisher method. In our implementation of the block Fisher method, both the gradient and the block-Fisher matrices were estimated with a moving-average scheme, with the decay factors being 0.9. In all of the other methods compared to the block Fisher method, moving averages were also used, with the decay factors being 0.9, as described in Section D in the Appendix, to compute the relevant gradients and approximate block-Fisher matrices used by them, based on values computed at points generated by the block-Fisher method.

As shown in Figure 1, the cosine similarity for TNT is always around 0.7 to 0.8, which is similar to (and sometimes higher) than the structure-aware method KFAC, and always better than Shampoo. To provide more information, we also include SGD with momentum and Adam, whose similarity to the block Fisher direction is usually lower that of the second-order methods.

## 4   Convergence

In this section, we present results on the convergence of an idealized version of TNT that uses the actual covariance of $\mathcal{D}\theta$, rather than a statistical estimate of it (see Algorithm 2 in the Appendix). In particular, our results show that Algorithm 2, with constant batch size and decreasing step size, converges to a stationary point under some mild assumptions. For simplicity, we assume that the model only contains one tensor variable $W$. However, our results can be easily extended to the case of multiple tensor variables. To start with, our proofs, which are delayed to Section B in the Appendix, require the following assumptions:

**Assumption 1.** $\mathcal{L} : \mathbb{R}^n \to \mathbb{R}$ *is continuously differentiable.* $\mathcal{L}(\theta)$ *is lower bounded by a real number* $\mathcal{L}^{low}$ *for any* $\theta \in \mathbb{R}^n$. $\nabla \mathcal{L}$ *is globally Lipschitz continuous with Lipschitz constant* $L$; *namely for any* $\theta, \theta' \in \mathbb{R}^n$, $\|\nabla\mathcal{L}(\theta) - \nabla\mathcal{L}(\theta')\| \leqslant L\|\theta - \theta'\|$.

**Assumption 2.** *For any iteration* $t$, *we have*

$$a) \ \mathbb{E}_{\xi_t}\left[\nabla l(\theta_t, \xi_t)\right] = \nabla\mathcal{L}(\theta_t) \qquad b) \ \mathbb{E}_{\xi_t}\left[\|\nabla l(\theta_t, \xi_t) - \nabla\mathcal{L}(\theta_t)\|^2\right] \leqslant \sigma^2$$

*where* $\sigma > 0$ *is the noise level of the gradient estimation, and* $\xi_t, t = 1, 2, \ldots,$ *are independent samples, and for a given* $t$ *the random variable* $\xi_t$ *is independent of* $\{\theta_j\}_{j=1}^t$

**Assumption 3.** *Let* $G := \mathcal{D}\theta$. *For any* $\theta \in \mathbb{R}^n$, *the norm of the Fisher matrix* $F = \mathbb{E}_{x \sim Q_x, y \sim p}[\overline{vec}(G)\overline{vec}(G)^\top]$ *is bounded above.*

Since $F$ represents the curvature of the KL divergence of the model's predictive distribution, Assumption 3 controls the change of predictive distribution when the model's parameters change; hence,

Table 1: Memory and per-iteration time complexity beyond that required by SGD

| Name | Memory | Time (per-iteration) |
|------|--------|----------------------|
| TNT | $O(\sum_{i=1}^{k} d_i^2)$ | $O((\frac{1}{T_1} m + \sum_{i=1}^{k} d_i) \prod_{i=1}^{k} d_i + \frac{1}{T_2} \sum_{i=1}^{k} d_i^3)$ |
| Shampoo | $O(\sum_{i=1}^{k} d_i^2)$ | $O((\sum_{i=1}^{k} d_i) \prod_{i=1}^{k} d_i + (\frac{1}{T_2} \sum_{i=1}^{k} d_i^3$- *if using SVD*$))$ |
| Adam-like | $O(\prod_{i=1}^{k} d_i)$ | $O(\prod_{i=1}^{k} d_i)$ |
| Newton-like | $O(\prod_{i=1}^{k} d_i^2)$ | *- depends on specific algorithm* |

it is a mild assumption for reasonable deep learning models. Essentially, we would like to prove that, if the Fisher matrix is upper bounded, our approximated Fisher (by TNT) is also upper bounded.

We now present two lemmas and our main theorem; see Section B in the Appendix for proofs.

**Lemma 1.** $\|\mathbb{E}_{x \sim Q_x, y \sim p}[G^{(i)}]\| \leqslant \left( \frac{1}{d_i} \prod_{i'=1}^{k} d_{i'} \right) \|\mathbb{E}_{x \sim Q_x, y \sim p}[\overline{vec}(G)\overline{vec}(G)^{\top}]\|, \ \forall \ i = 1, \ldots, k.$

**Lemma 2.** *Suppose Assumption 3 holds. Let* $F_{TNT} := (\tilde{U}_1 + \epsilon I) \otimes \cdots \otimes (\tilde{U}_k + \epsilon I)$ *where* $\tilde{U}_i$*'s are defined in (5). Then, the norm of* $F_{TNT}$ *is bounded both above and below.*

**Theorem 1.** *Suppose that Assumptions 1, 2, and 3 hold for* $\{\theta_t\}$ *generated by Algorithm 2 with batch size* $m_t = m$ *for all t. If we choose* $\alpha_t = \frac{\underline{\kappa}}{L\bar{\kappa}^2} t^{-\beta}$*, with* $\beta \in (0.5, 1)$*, then*

$$\frac{1}{N} \sum_{t=1}^{N} \mathbb{E}_{\{\xi_j\}_{j=1}^{\infty}} \left[ \|\nabla\mathcal{L}(\theta_t)\|^2 \right] \leqslant \frac{2L \left( M_\mathcal{L} - \mathcal{L}^{low} \right) \bar{\kappa}^2}{\underline{\kappa}^2} N^{\beta-1} + \frac{\sigma^2}{(1-\beta)m} \left( N^{-\beta} - N^{-1} \right),$$

*where* $N$ *denotes the iteration number and the constant* $M_\mathcal{L} > 0$ *depends only on* $\mathcal{L}$*. Moreover, for a given* $\delta \in (0, 1)$*, to guarantee that* $\frac{1}{N} \sum_{t=1}^{N} \mathbb{E}_{\{\xi_j\}_{j=1}^{\infty}} \left[ \|\nabla\mathcal{L}(\theta_t)\|^2 \right] < \delta$*, $N$ needs to be at most* $O \left( \delta^{-\frac{1}{1-\beta}} \right).$

## 5  Implementation Details of TNT and Comparison on Complexity

**Implementation Details of TNT.** In practice, we compute $\overline{G} = \overline{\mathcal{D}W}$ averaged over a minibatch of data at every iteration, and use the value of $\overline{G}^{(i)}$ to update a moving average estimate $\widehat{G^{(i)}}$ of $\mathbb{E}[G^{(i)}]$. The extra work for these computations (as well as for updating the inverses of $\tilde{U}_i$) compared with a stochastic gradient descent method is amortized by only performing them every $T_1$ (and $T_2$) iterations, which is also the approach used in KFAC and Shampoo, and does not seems to degrade the overall performance of the TNT algorithm. Moreover, we compute $\mathbb{E}[G^{(i)}]$ using the whole dataset at the initialization point as a warm start, which is also done in our implementations of Shampoo and KFAC. See Algorithm 3 in the Appendix for the detailed implementation of TNT.

**A Comparison on Memory and Per-iteration Time Complexity.** To compare the memory requirements and per-iteration time complexities of different methods, we consider the case where we optimize one tensor variable of size $d_1 \times \cdots \times d_k$ using minibatches of size $m$ at every iteration. A plain SGD method requires $O(\prod_{i=1}^{k} d_i)$ to store the model parameters and the gradient, whereas its per-iteration time complexity is $O(m \prod_{i=1}^{k} d_i)$. Table 1 lists the memory requirements and per-iteration time complexities in excess of that required by SGD for different methods.

Compared with a classic Newton-like method (e.g. BFGS), TNT (as well as Shampoo) reduces the memory requirement from $O(\prod_{i=1}^{k} d_i^2)$ to $O(\sum_{i=1}^{k} d_i^2)$, which is comparable to that of Adam-like adaptive gradient methods. In fact, if the $d_i$'s are all equal to $d$ and $3 \leqslant k << d$, the Kronecker-factored TNT pre-conditioning matrix requires $kd^2$ storage, which is less than that required by the diagonal pre-conditioners used by Adam-like methods. On the other hand, in terms of per-iteration time complexity, TNT (as well as Shampoo) only introduces a mild overhead for estimating the statistics $\mathbb{E}[G^{(i)}]$'s, inverting the pre-conditioning matrices, and computing the updating direction. Also, the first two of these operations can be amortized by only performing them every $T_1$ and $T_2$ iterations. Lastly, the extra work of $O(\frac{1}{T_1} m \prod_{i=1}^{k} d_i)$ required by TNT relative to Shampoo is due to the extra backward pass needed to estimate the true Fisher, as opposed to the EF.

Moreover, although TNT and Shampoo both incur $\frac{1}{T_2} \sum_{i=1}^{k} d_i^3$ amortized time to invert the preconditioning matrices, the SVD operation in Shampoo can take much more time than the matrix inverse operation in TNT, especially when the matrix size is large[2].

The per-iteration computational complexity of KFAC is more complicated because it depends on the type of the layer/variable. For linear layers, TNT and KFAC both uses two matrices, whose sizes are the number of input nodes and output nodes, respectively. For convolutional layers, TNT uses three matrices, whose sizes are the size of filter, number of input channels, and number of output channels, whereas KFAC uses two matrices whose sizes are the size of filter times number of input channels, and number of output channels. As a result, the first KFAC matrix requires much more memory. In general, the per-iteration complexity of KFAC is no less than that of TNT.

## 6 Experiments

In this section, we compare TNT with some state-of-the-art second-order (KFAC, Shampoo) and first-order (SGD with momentum, Adam) methods (see Section D.1 in the Appendix on how these methods were implemented). The Hessian-based K-BFGS method [15, 38] is another state-of-the-art Kronecker-factored second-order method for training deep learning models. Since our focus is on optimizers that use Fisher or empirical Fisher as the preconditioning matrix, we did not include K-BFGS in our comparison.

Our experiments were run on a machine with one V100 GPU and eight Xeon Gold 6248 CPUs using PyTorch [37]. Each algorithm was run using the best hyper-parameters, determined by an appropriate grid search (specified below), and 5 different random seeds. In Figures 2 and 3 the performance of each algorithm is plotted: the solid curves give results obtained by averaging the 5 different runs, and the shaded area depicts the ±standard deviation range for these runs. Our code is available at https://github.com/renyiryry/tnt_neurips_2021.

### 6.1 Optimization: Autoencoder Problems

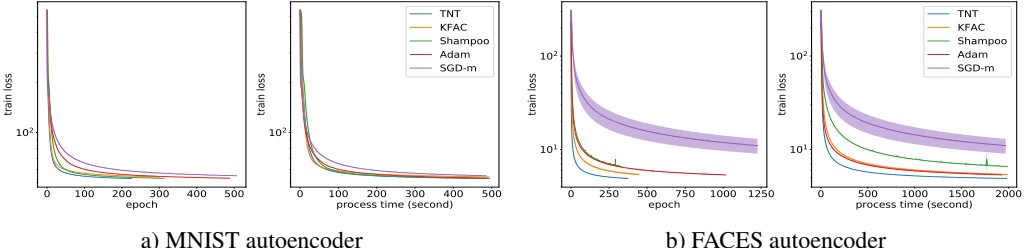

a) MNIST autoencoder               b) FACES autoencoder

Figure 2: Optimization performance of TNT, KFAC, Shampoo, Adam, and SGD-m on two autoencoder problems

We first compared the optimization performance of each algorithm on two autoencoder problems [21] with datasets MNIST [27] and FACES[3], which were also used in [32, 34, 5, 15] as benchmarks to compare different algorithms. For each algorithm, we conducted a grid search on the learning rate and damping value based on the criteria of minimal training loss. We set the Fisher matrix update frequency $T_1 = 1$ and inverse update frequency $T_2 = 20$ for all of the second-order methods. Details of our experiment settings are listed in Section D.2 in the Appendix. From Figure 2, it is clear that TNT outperformed SGD with momentum and Adam, both in terms of per-epoch progress and process time. Moreover, TNT performed (at least) as well as KFAC and Shampoo, with a particularly strong performance on the FACES dataset. We repeated these experiments using a grid search on more hyper-parameters, and obtained results (see Figure 6 in Sec D.5) that further support our observations based on Figure 2.

---

[2]In [3] it is shown that replacing the SVD operation by a coupled Schur-Newton method saves time for matrices of size greater than $1000 \times 1000$. In our experiments, we used the coupled Newton method implementation of Shampoo.

[3]https://cs.nyu.edu/~roweis/data.html

## 6.2 Generalization: Convolutional Neural Networks

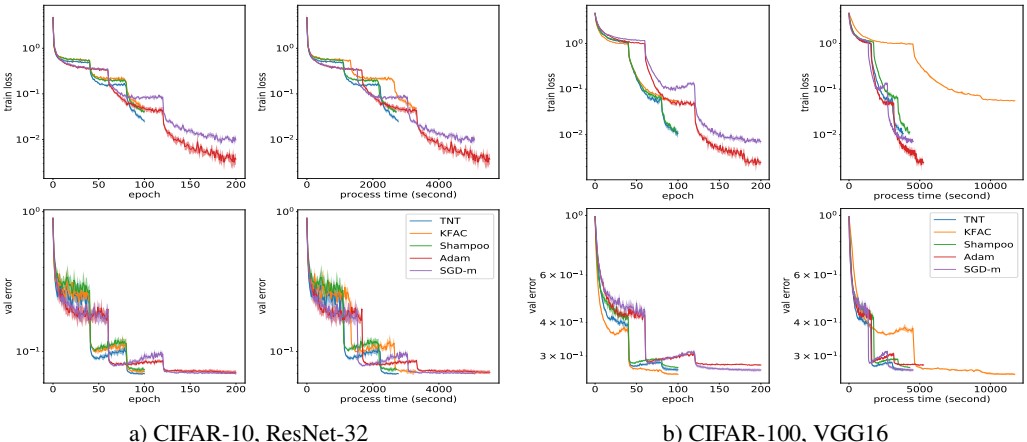

a) CIFAR-10, ResNet-32                 b) CIFAR-100, VGG16

Figure 3: Generalization ability of TNT, KFAC, Shampoo, Adam, and SGD-m on two CNN models. Upper row depicts the training loss whereas lower row depicts the validation classification error.

We then compared the generalization ability of each algorithm on two CNN models, namely, ResNet32 [19] (with CIFAR10 dataset [24]) and VGG16 [42] (with CIFAR100 dataset [24]). The first-order methods were run for 200 epochs during which the learning rate was decayed by a factor of 0.1 every 60 epochs, whereas the second-order methods were run for 100 epochs during which the learning rate was decayed by a factor of 0.1 every 40 epochs; (these settings are the same as in [45]). Moreover, as indicated in [29, 45], weight decay, different from the $L_2$ regularization added to the loss function, helps improve generalization across different optimizers. Thus, for each algorithm, we conducted a grid search on the initial learning rate and the weight decay factor based on the criteria of maximal validation classification accuracy. The damping parameter was set to 1e-8 for Adam (following common practice), and 0.03 for KFAC[4]. For TNT and Shampoo, we set $\epsilon = 0.01$. We set $T_1 = 10$ and $T_2 = 100$ for the three second-order methods (same as in [45]). Details of our experiment settings and a further discussion of the choice of damping hyper-parameters can be found in Section D.3 in the Appendix.

The results in Figure 3 indicate that, with a proper learning rate and weight decay factor, second-order methods and Adam exhibit roughly the same generalization performance as SGD with momentum, which corroborate the results in [29, 45]. In particular, TNT has a similar (and sometimes better) generalization performance than the other methods. For example, comparing TNT with SGD-m, TNT (SGD-m) achieves 93.08% (93.06%) validation accuracy with ResNet32 on CIFAR10 and 73.33% (73.43%) validation accuracy with VGG16 on CIFAR-100, after 100 (200) epochs (see Table 3 in the Appendix for the accuracy achieved by the other algorithms). Moreover, in terms of process time, TNT is roughly twice (equally) as fast as SGD with momentum on ResNet32/CIFAR10 in Figure 3a (on VGG16 on CIFAR-100 in Figure 3b). This illustrates the fact that TNT usually requires only moderately more computational effort per-iteration but fewer iterations to converge than first-order methods. Also, as shown on the VGG16 model, KFAC seems to be much slower than TNT and Shampoo on larger models. This is because the most recent version of KFAC, which we implemented, uses SVD (i.e., eigenvalue decomposition) to compute inverse matrices (see Section D.1.2 in the Appendix for a discussion of this). In contrast, TNT does not need to use SVD, and the most recent version of Shampoo replaces SVD with a coupled Newton method in [3].

We also compared TNT with a variant of it that uses the empirical rather than the true Fisher as the preconditioning matrix. The results of this comparison, which are presented in Section D.4 in the Appendix, suggest that it is preferable to use Fisher rather than empirical Fisher as pre-conditioning matrices in TNT.

---

[4]The value of 0.03 is suggested in `https://github.com/alecwangcq/KFAC-Pytorch`, a github repo by the authors of [45].

# 7 Conclusion and Further Discussions

In this paper, we proposed a new second-order method, and in particular, an approximate natural gradient method TNT, for training deep learning models. By approximating the Fisher matrix using the structure imposed by the tensor normal distribution, TNT only requires mild memory and computational overhead compared with first-order methods. Our experiments on various deep learning models and datasets, demonstrate that TNT provides comparable and sometimes better results than the state-of-the-art (SOTA) methods, both from the optimization and generalization perspectives.

Due to space and computational resource constraints, we did not run experiments on even larger models such as ImageNet and advanced models for NLP tasks. However, the results in this paper already show very strong evidence of the potential of the TNT method. We also did not explore extending our method to a distributed setting, which has been shown to be a promising direction for second-order methods such as KFAC and Shampoo [4, 3]. Since TNT already performs very well on a single machine, we expect that it will continue to do so in a distributed setting. These issues will be addressed in future research. We did not compare TNT with the SOTA Kronecker-based quasi-Newton methods [15, 38], since they are not as closely related to TNT as are Shampoo and KFAC. Their performance relative to TNT can be inferred from the comparisons here combined with those reported in [15, 38, 3].

As a final note[5], the preconditioning matrices of TNT (as well as those of Shampoo) are derived from the specific shape of the (tensor) parameters of the particular deep learning model that is being trained. One can, of course, reshape these parameters, e.g., by flattening the tensors into vectors, which gives rise to very different preconditioning matrices.

The method proposed in this paper can be applied to any deep learning or machine learning model. If the model and/or data has a flawed design or contains bias, this could potentially have negative societal impacts. However, this possibility is beyond the scope of the work presented in this paper.

---

[5]We thank the program chair for pointing this out.

## Acknowledgments and Disclosure of Funding

We would like to thank the anonymous reviewers for their very helpful comments and suggestions.

The research efforts of D. Goldfarb and Y. Ren on this paper were supported in part by NSF Grant IIS-1838061.

We acknowledge computing resources from Columbia University's Shared Research Computing Facility project, which is supported by NIH Research Facility Improvement Grant 1G20RR030893-01, and associated funds from the New York State Empire State Development, Division of Science Technology and Innovation (NYSTAR) Contract C090171, both awarded April 15, 2010.

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
