# A  Some Tensor Definitions and Properties

We present in this section fairly standard notation and definitions regarding tensors, e.g., see [18] and Chapter 3 of [30], that we use throughout the paper. Let $A \in \mathbb{R}^{d_1 \times \cdots \times d_k}$ denote a tensor of order $k$.

- *slices* of $A$ along its $i$-th dimension: for $i = 1, ..., k$ and $j = 1, ..., d_i$, the $j$-th slice of $A$ along its $i$-th dimension, $A_j^i$ denotes the $d_1 \times \cdots \times d_{i-1} \times d_{i+1} \times \cdots \times d_k$ tensor of order $k - 1$, composed from all of the entries of $A$ whose $i$th index is $j$.

- *vectorization* of $A$: denoted as $\overline{\mathrm{vec}}(A)$, is defined recursively as

$$\overline{\mathrm{vec}}(A) = \begin{pmatrix} \overline{\mathrm{vec}}(A_1^1) \\ \vdots \\ \overline{\mathrm{vec}}(A_{d_1}^1) \end{pmatrix},$$

  where for the base case, in which $A$ is one-dimensional tensor (i.e., a vector), $\overline{\mathrm{vec}}(A) = A$. Note that when $A$ is a matrix, this corresponds to the row-major vectorization of $A$.

- *matricization* of $A$: denoted as $\mathrm{mat}_i(A)$, for $i = 1, ..., k$, is defined as

$$\mathrm{mat}_i(A) = \begin{pmatrix} \overline{\mathrm{vec}}(A_1^i)^\top \\ \vdots \\ \overline{\mathrm{vec}}(A_{d_i}^i)^\top \end{pmatrix}.$$

  Note that $\overline{\mathrm{vec}}(A) = \overline{\mathrm{vec}}(\mathrm{mat}_1(A))$.

- *contraction* of $A$ with itself along all but the $i$th dimension: denoted as $A^{(i)}$, is defined as $\mathrm{mat}_i(A)\mathrm{mat}_i(A)^\top$.

- *mode-$i$ product* of $A$ by a matrix $U \in \mathbb{R}^{d_i' \times d_i}$: the operation is denoted as $\times_i$. Let $B = A \times_i U \in \mathbb{R}^{d_1 \times \cdots \times d_{i-1} \times d_i' \times d_{i+1} \times \cdots \times d_k}$ denote the resulting tensor. $B_{j_1, ..., j_{i-1}, j_i', j_{i+1}, ..., j_k} = \sum_{j_i} A_{j_1, ..., j_k} U_{j_i', j_i}$. Note that in the matrix case ($k = 2$), $A \times_1 U = UA$, $A \times_2 U = AU^\top$.

**Lemma 3.** *Let $X \in \mathbb{R}^{m \times n}$, $A \in \mathbb{R}^{m \times m}$, $B \in \mathbb{R}^{n \times n}$. Then, we have*

$$(A \otimes B^\top)\overline{vec}(X) = \overline{vec}(AXB).$$

Note that the above lemma is slightly different from the most common version of it, which uses a column-major vectorization of the matrix $X$.

**Proposition 1.** *Let $G \in \mathbb{R}^{d_1 \times \cdots \times d_k}$ and $U_i \in \mathbb{R}^{d_i \times d_i}$ for $i = 1, ..., k$. Then, we have*

$$\left(\otimes_{i=1}^k U_i\right) \overline{vec}(G) = \overline{vec}(G \times_1 U_1 \times_2 U_2 \cdots \times_k U_k). \tag{6}$$

**Proof of Proposition 1:**

*Proof.* Our proof, which is largely inspired by the one in [18], is by induction on $k$. When $k = 1$, it is easy to see that (6) holds by the definition of the mode-$i$ product. When $k = 2$, (6) follows from Lemma 3.

Now assume that (6) holds for $1, 2, ..., k - 1$. For $k$, we let $H = \otimes_{i=2}^k U_i$

By the induction hypothesis,

$$\mathrm{mat}_1(G)H^\top = \left(H\mathrm{mat}_1(G)^\top\right)^\top = \left(H\left(\overline{\mathrm{vec}}(G_1^1) \quad \cdots \quad \overline{\mathrm{vec}}(G_{d_1}^1)\right)\right)^\top \tag{7}$$

$$= \left(H\overline{\mathrm{vec}}(G_1^1) \quad \cdots \quad H\overline{\mathrm{vec}}(G_{d_1}^1)\right)^\top \tag{8}$$

$$= \left(\overline{\mathrm{vec}}(G_1^1 \times_1 U_2 \cdots \times_{k-1} U_k) \quad \cdots \quad \overline{\mathrm{vec}}(G_{d_1}^1 \times_1 U_2 \cdots \times_{k-1} U_k)\right)^\top \tag{9}$$

$$= \begin{pmatrix} \overline{\mathrm{vec}}(G_1^1 \times_1 U_2 \cdots \times_{k-1} U_k)^\top \\ \vdots \\ \overline{\mathrm{vec}}(G_{d_1}^1 \times_1 U_2 \cdots \times_{k-1} U_k)^\top \end{pmatrix} = \mathrm{mat}_1(G \times_2 U_2 \cdots \times_k U_k) \tag{10}$$

By Lemma 3 and (10),

$$
\begin{aligned}
\left(\otimes_{i=1}^{k} U_i\right)\overline{\mathrm{vec}}(G) &= (U_1 \otimes H)\,\overline{\mathrm{vec}}(\mathrm{mat}_1(G)) = \overline{\mathrm{vec}}(U_1\mathrm{mat}_1(G)H^\top) \\
&= \overline{\mathrm{vec}}(U_1\mathrm{mat}_1(G \times_2 U_2 \cdots \times_k U_k)) \\
&= \overline{\mathrm{vec}}(\mathrm{mat}_1(G \times_2 U_2 \cdots \times_k U_k \times_1 U_1)) \\
&= \overline{\mathrm{vec}}(G \times_2 U_2 \cdots \times_k U_k \times_1 U_1) \\
&= \overline{\mathrm{vec}}(G \times_1 U_1 \times_2 U_2 \cdots \times_k U_k),
\end{aligned}
$$

where the third from last equality comes from the fact that $B\mathrm{mat}_i(A) = \mathrm{mat}_i(A \times_i B)$, and the last equality comes from the fact that mode-$i$ products are commutative.

$\square$

## B   Proofs of Lemmas and Theorem 1

---
**Algorithm 2** Idealized Version of TNT

---
**Require:** Given $\theta_1 \in \mathbb{R}^n$, batch sizes $\{m_t\}_{t \geqslant 1}$, step sizes $\{\alpha_t\}_{t \geqslant 1}$, and damping value $\epsilon > 0$
1: **for** $t = 1, 2, \ldots$ **do**
2:    Sample mini-batch of size $m_t$: $M_t = \{\xi_{t,i}, i = 1, \ldots, m_t\}$
3:    Calculate $\widehat{\nabla \mathcal{L}}_t = \frac{1}{m_t}\sum_{\xi_{t,i} \in M_t}\nabla l(\theta_t, \xi_{t,i})$
4:    Compute $\tilde{U}_i$ $(i = 1, ..., k)$ by formula (5), using the **true values** of $\mathbb{E}_{x \sim Q_x, y \sim p}[G^{(i)}]$ $(i = 1, ..., k)$ at the current parameter $\theta_t$.
5:    Compute $p_t = \overline{\mathrm{vec}}\left(\widehat{\nabla \mathcal{L}}_t \times_1 (\tilde{U}_1 + \epsilon I)^{-1} \times_2 \cdots \times_k (\tilde{U}_k + \epsilon I)^{-1}\right)$
6:    Calculate $\theta_{t+1} = \theta_t - \alpha_t p_t$
7: **end for**

---

Algorithm 2 describes an idealized version of TNT, whose convergence is verified by the proofs of Lemmas 1 and 2, and Theorem 1 below.

**Lemma 1.** $\|\mathbb{E}_{x \sim Q_x, y \sim p}[G^{(i)}]\| \leqslant \left(\frac{1}{d_i}\prod_{i'=1}^{k} d_{i'}\right)\|\mathbb{E}_{x \sim Q_x, y \sim p}[\overline{vec}(G)\overline{vec}(G)^\top]\|$, $\forall\, i = 1, \ldots, k$.

**Proof of Lemma 1:**

*Proof.* Let $X \in \mathbb{R}^{m \times n}$ be a random matrix, and $x_i \in \mathbb{R}^m$ denote its $i$th column $(i = 1, ..., n)$. Because $\overline{\mathrm{vec}}(X)$ is a vector containing all the elements of all the $x_i$'s, $x_i x_i^\top$ is a square submatrix of $\overline{\mathrm{vec}}(X)\overline{\mathrm{vec}}(X)^\top$. Hence, $\|\mathbb{E}[x_i x_i^\top]\| \leqslant \|\mathbb{E}[\overline{\mathrm{vec}}(X)\overline{\mathrm{vec}}(X)^\top]\|$, and we have that

$$
\|\mathbb{E}[XX^\top]\| = \|\mathbb{E}[\sum_{i=1}^{n} x_i x_i^\top]\| = \|\sum_{i=1}^{n}\mathbb{E}[x_i x_i^\top]\| \leqslant \sum_{i=1}^{n}\|\mathbb{E}[x_i x_i^\top]\| \leqslant n\|\mathbb{E}[\overline{\mathrm{vec}}(X)\overline{\mathrm{vec}}(X)^\top]\|.
$$

Letting $X = \mathrm{mat}_i(G) \in \mathbb{R}^{d_i \times (d_1 \cdots d_{i-1} d_{i+1} \cdots d_k)}$, it then follows that

$$
\begin{aligned}
\|\mathbb{E}_{x \sim Q_x, y \sim p}[G^{(i)}]\| &\leqslant (d_1 \cdots d_{i-1}d_{i+1} \cdots d_k)\|\mathbb{E}[\overline{\mathrm{vec}}(\mathrm{mat}_i(G))\overline{\mathrm{vec}}(\mathrm{mat}_i(G))^\top]\| \\
&= (\frac{1}{d_i}\prod_{i'=1}^{k} d_{i}')\|\mathbb{E}[\overline{\mathrm{vec}}(G)\overline{\mathrm{vec}}(G)^\top]\|.
\end{aligned}
$$

$\square$

**Lemma 2.** *Suppose Assumption 3 holds. Let $F_{TNT} := (\tilde{U}_1 + \epsilon I) \otimes \cdots \otimes (\tilde{U}_k + \epsilon I)$, where the $\tilde{U}_i$'s are defined in (5). Then, the norm of $F_{TNT}$ is bounded both above and below.*

**Proof of Lemma 2:**

*Proof.* It is clear that $||F_{\text{TNT}}|| = \prod_{i=1}^{k} ||\tilde{U}_i + \epsilon I|| \geqslant \epsilon^k$. On the other hand, for $i = 1, ..., k$, if we denote the eigenvalues of $\mathbb{E}[G^{(i)}]$ by $\lambda_1 \leqslant \cdots \leqslant \lambda_{d_i}$, we have from (5) that

$$\|\tilde{U}_i\| = \frac{\|\mathbb{E}[G^{(i)}]\|}{\left(\frac{\text{tr}(\mathbb{E}[G^{(i)}])}{\prod_j d_j}\right)^{(k-1)/k} \prod_{j \neq i} d_j} = \frac{\lambda_{d_i}}{\left(\frac{\lambda_1 + \cdots + \lambda_{d_i}}{\prod_j d_j}\right)^{(k-1)/k} \prod_{j \neq i} d_j}$$

$$\leqslant \frac{\lambda_{d_i}}{\left(\frac{\lambda_{d_i}}{\prod_j d_j}\right)^{(k-1)/k} \prod_{j \neq i} d_j} = \frac{d_i \lambda_{d_i}^{1/k}}{(\prod_j d_j)^{1/k}} = \frac{d_i \|\mathbb{E}[G^{(i)}]\|^{1/k}}{(\prod_j d_j)^{1/k}}.$$

Thus, since $||F_{\text{TNT}}|| = \prod_{i=1}^{k} ||\tilde{U}_i + \epsilon I|| = \prod_{i=1}^{k}(||\tilde{U}_i|| + \epsilon)$, by the above and Lemma 1,

$$||F_{\text{TNT}}|| \leqslant \prod_{i=1}^{k} \left(\frac{d_i \|\mathbb{E}[G^{(i)}]\|^{1/k}}{(\prod_j d_j)^{1/k}} + \epsilon\right) \leqslant \prod_{i=1}^{k} \left(d_i^{1-1/k} ||\mathbb{E}[\overline{\text{vec}}(G)\overline{\text{vec}}(G)^\top]||^{1/k} + \epsilon\right).$$

Then, by Assumption 3, we have that $||F_{\text{TNT}}||$ is bounded above.

$\square$

**Proof of Theorem 1:**

*Proof.* The proof of Theorem 1 follows from Theorem 2.8 in [44]. Clearly, Algorithm 2 falls under the scope of the stochastic quasi-Newton (SQN) method in [44]. In particular, by Proposition 1, the pre-conditioning matrix $H = F_{\text{TNT}}^{-1}$. Moreover, to apply Theorem 2.8 in [44], we need to show that AS.1 - AS.4 in [44] hold. First, AS.1 and AS.2 in [44] are the same as Assumption 1 and Assumption 2, respectively in Section 4 in our paper. Second, by Lemma 2, since $||F_{\text{TNT}}||$ is both upper and lower bounded, so is $||F_{\text{TNT}}^{-1}||$. Hence, AS.3 in [44] is ensured. Finally, Algorithm 2 itself ensures AS.4 in [44] holds. Hence, by Theorem 2.8 of [44], the result is guaranteed.

$\square$

## C  Pseudo-code for TNT

In Algorithm 3, we present a detailed pseudo-code for our actual implementation of TNT. The highlighted parts, i.e., Lines 7, 15 and 16, indicate where TNT differs significantly from Shampoo.

## D  Details of the Experiments

In our implementations of the algorithms that we compared to TNT, we included in all of the techniques like weight decay and momentum, so that our numerical experiments would provide a FAIR comparison. Consequently, we did not include some special techniques that have been incorporated in some of the algorithms as described in previously published papers, since to keep the comparisons fair, we would have had to incorporate such techniques in all of the algorithms (see Section D.1.1 for more details).

### D.1  Competing Algorithms

In SGD with momentum, we updated the momentum of the gradient $m = \mu \cdot m + g$ at every iteration, where $g$ denotes the minibatch gradient at current iteration. The gradient momentum is also used in the second-order methods, in our implementations.

For Adam, we follow exactly the algorithm in [23] with $\beta_1 = 0.9$ and $\beta_2 = 0.999$. In particular, we follow the approach in [23] in estimating the momentum of gradient by $m = \beta_1 \cdot m + (1 - \beta_1) \cdot g$. The role of $\beta_1$ and $\beta_2$ is similar to that of $\mu$ and $\beta$ in Algorithm 3 and Algorithm 4, as we will describe below.

In the experiments on CNNs, we use weight decay (same as in Algorithms 3 and 4) on SGD and Adam, similar to SGDW and AdamW in [29] (for further details, see Section D.3).

**Algorithm 3** Tensor-Normal Training

**Require:** Given batch size $m$, learning rate $\{\alpha_t\}_{t \geqslant 1}$, weight decay factor $\gamma$, damping value $\epsilon$, statistics update frequency $T_1$, inverse update frequency $T_2$

1: $\mu = 0.9, \beta = 0.9$
2: Initialize $\widehat{G_l^{(i)}} = \mathbb{E}[G_l^{(i)}]$ ($l = 1, .., k$, $i = 1, ..., k_l$) by iterating through the whole dataset, $\widehat{\nabla_{W_l}\mathcal{L}} = 0$ ($l = 1, ..., L$)
3: **for** $t = 1, 2, \dots$ **do**
4:      Sample mini-batch $M_t$ of size $m$
5:      Perform a forward-backward pass over $M_t$ to compute the mini-batch gradient $\overline{\nabla\mathcal{L}}$
6:      **if** $t \equiv 0 \pmod{T_1}$ **then**
7:          Perform another backward pass over $M_t$ with $y$ sampled from the predictive distribution to compute $\overline{G_l} = \overline{\mathcal{D}W_l}$ averaged across $M_t$ ($l = 1, ..., L$)
8:      **end if**
9:      **for** $l = 1, ...L$ **do**
10:          $\widehat{\nabla_{W_l}\mathcal{L}} = \mu\widehat{\nabla_{W_l}\mathcal{L}} + \overline{\nabla_{W_l}\mathcal{L}}$
11:          **if** $t \equiv 0 \pmod{T_1}$ **then**
12:             Update $\widehat{G_l^{(i)}} = \beta\widehat{G_l^{(i)}} + (1-\beta)\overline{G_l}^{(i)}$ for $i = 1, ..., k_l$
13:          **end if**
14:          **if** $t \equiv 0 \pmod{T_2}$ **then**
15:             Determine $\tilde{U}_1^{(l)}, ..., \tilde{U}_{k_l}^{(l)}$ from $\widehat{G_l^{(1)}}, ..., \widehat{G_l^{(k_l)}}$ by (5)
16:             Recompute $(\tilde{U}_1^{(l)} + \epsilon I)^{-1}, ..., (\tilde{U}_{k_l}^{(l)} + \epsilon I)^{-1}$
17:          **end if**
18:          $p_l = \widehat{\nabla_{W_l}\mathcal{L}} \times_1 (\tilde{U}_1^{(l)} + \epsilon I)^{-1} \times_2 \cdots \times_k (\tilde{U}_k^{(l)} + \epsilon I)^{-1}$
19:          $p_l = p_l + \gamma W_l$
20:          $W_l = W_l - \alpha_t \cdot p_l$.
21:      **end for**
22: **end for**

### D.1.1 Shampoo

---

**Algorithm 4** Shampoo

---

**Require:** Given batch size $m$, learning rate $\{\alpha_t\}_{t \geqslant 1}$, weight decay factor $\gamma$, damping value $\epsilon$, statistics update frequency $T_1$, inverse update frequency $T_2$

1: $\mu = 0.9, \beta = 0.9$
2: Initialize $\widehat{G_l^{(i)}} = \mathbb{E}[G_l^{(i)}]$ ($l = 1, .., k$, $i = 1, ..., k_l$) by iterating through the whole dataset, $\widehat{\nabla_{W_l}\mathcal{L}} = 0$ ($l = 1, ..., L$)
3: **for** $t = 1, 2, \ldots$ **do**
4:     Sample mini-batch $M_t$ of size $m$
5:     Perform a forward-backward pass over the current mini-batch $M_t$ to compute the minibatch gradient $\overline{\nabla\mathcal{L}}$
6:     **for** $l = 1, ...L$ **do**
7:         $\widehat{\nabla_{W_l}\mathcal{L}} = \mu\widehat{\nabla_{W_l}\mathcal{L}} + \overline{\nabla_{W_l}\mathcal{L}}$
8:         **if** $t \equiv 0 \pmod{T_1}$ **then**
9:             Update $\widehat{G_l^{(i)}} = \beta\widehat{G_l^{(i)}} + (1 - \beta)\overline{G_l}^{(i)}$ for $i = 1, ..., k_l$ where $\overline{G_l} = \overline{\nabla_{W_l}\mathcal{L}}$
10:         **end if**
11:         **if** $t \equiv 0 \pmod{T_2}$ **then**
12:             Recompute $\left(\widehat{G_l^{(1)}} + \epsilon I\right)^{-1/2k_l}, ..., \left(\widehat{G_l^{(k_l)}} + \epsilon I\right)^{-1/2k_l}$ with the coupled Newton method
13:         **end if**
14:         $p_l = \widehat{\nabla_{W_l}\mathcal{L}} \times_1 \left(\widehat{G_l^{(1)}} + \epsilon I\right)^{-1/2k_l} \times_2 \cdots \times_k \left(\widehat{G_l^{(k_l)}} + \epsilon I\right)^{-1/2k_l}$
15:         $p_l = p_l + \gamma W_l$
16:         $W_l = W_l - \alpha_t \cdot p_l$
17:     **end for**
18: **end for**

---

In Algorithm 4, we present our implementation of Shampoo, which mostly follows the description of it given in [18]. Several major improvements are also included, following the suggestions in [3], including:

1. In Line 9 of Algorithm 4, a moving average is used to update the estimates $\widehat{G_l^{(i)}}$, as is done in our implementations of TNT and KFAC. This approach is also used in Adam, whereas summing the $G_l^{(i)}$'s over all iterations, as in [18], is analogous to what is done in AdaGrad, upon which Shampoo is based.

2. In Line 12 of Algorithm 4, we use a coupled Newton method to compute inverse roots of the matrices (as proposed in [3]), rather than using SVD. The coupled Newton approach has been shown to be much faster than SVD, and also preserves relatively good accuracy in terms of computing inverse roots. The coupled Newton method performs reasonably well (without tuning) using a max iteration number of 100 and an error tolerance of 1e-6.

Some other modifications proposed in [3] are not included in our implementation of Shampoo, mainly because these modifications can also be applied to TNT, and including them only in Shampoo would introduce other confounding factors.

(i) We did not explore multiplying the damping term in the pre-conditioner by the maximum eigenvalue $\lambda_{max}$ of the contraction matrix. Moreover, this modification is somewhat problematic, since, if the model contains any variables that always have a zero gradient (e.g. the bias in a convolutional layer that is followed by a BN layer), the optimizer would become unstable because the pre-conditioner of the zero-gradient variables would be the zero matrix, (note that in this case $\lambda_{max} = 0$).

(ii) We did not explore the diagonal variant of Shampoo, as we mainly focused on the comparison between different pre-conditioning matrices, and TNT can also be extended to a diagonal

version; similarly, we did not explore the variant proposed in [3] that divides large tensors into small blocks.

### D.1.2 KFAC

In this subsection, we briefly describe our implementation of KFAC. The preconditioning matrices that we used for linear layers and convolutional layers are precisely as those described in [34] and [17], respectively. For the parameters in the BN layers, we used the gradient direction, exactly as in https://github.com/alecwangcq/KFAC-Pytorch.

As in our implementations of TNT and Shampoo, and as suggested in [17], we did a warm start to estimate the pre-conditioning KFAC matrices in an initialization step that iterated through the whole data set, and adopted a moving average scheme to update them with $\beta = 0.9$ afterwards.

In inverting the KFAC matrices and computing the updating direction,

- for the autoencoder experiments, we inverted the damped KFAC matrices and used them to compute the updating direction, where the damping factors for both $A$ and $G$ were set to be $\sqrt{\lambda}$, where $\lambda$ is the overall damping value;[6]

- for the CNN experiments, we followed the SVD (i.e. eigenvalue decomposition) implementation suggested in https://github.com/alecwangcq/KFAC-Pytorch, which, as we verified, performs better than splitting the damping value and inverting the damped KFAC matrices (as suggested in [34, 17]).

Further, we implemented weight decay exactly as in TNT (Algorithm 3) and Shampoo (Algorithm 4).

### D.2 Experiment Settings for the Autoencoder Problems

Table 2: Hyper-parameters (learning rate, damping) used to produce Figure 2

| Name | MNIST | FACES |
| --- | --- | --- |
| TNT | (1e-4, 0.1) | (1e-6, 0.003) |
| KFAC | (0.003, 0.3) | (0.1, 10) |
| Shampoo | (3e-4, 3e-4) | (3e-4, 3e-4) |
| Adam | (1e-4, 1e-4) | (1e-4, 1e-4) |
| SGD-m | (0.003, -) | (0.001, -) |

MNIST has 60,000 training data, whereas FACES[7] has 103,500 training data. For all algorithms, we used a batch size of 1,000 at every iteration.

The autoencoder model used for MNIST has layer widths 784-1000-500-250-30-250-500-1000-784 with ReLU activation functions, except for the middle layer which uses a linear function and the last layer which uses a sigmoid function. The autoencoder model used for FACES has layer widths 625-2000-1000-500-30-500-1000-2000-625 with ReLU activation functions, except for the middle and last layers which use linear functions. We used binary entropy loss for MNIST and squared error loss for FACES. The above settings largely mimic the settings in [32, 34, 5, 15]. Since we primarily focused on optimization rather than generalization in these tasks, we did not include $L_2$ regularization or weight decay.

In order to obtain Figure 2, we first conducted a grid search on the learning rate (lr) and damping value based on the criteria of minimizing the training loss. The ranges of the grid searches used for the algorithms in our tests were:

- SGD-m:
  - lr: 1e-4, 3e-4, 0.001, 0.003, 0.01, 0.03

---

[6]Note that there are more sophisticated ways of splitting the damping value, such as one that makes use of the norms of the undamped matrices, to enforce that the two matrices have the same norm. See [34] and [17] for more on this.

[7]Downloadable at www.cs.toronto.edu/~jmartens/newfaces_rot_single.mat.

- – damping: not applicable
- Adam:
  - – lr: 1e-5, 3e-5, 1e-4, 3e-4, 0.001, 0.003, 0.01
  - – damping (i.e. the $\epsilon$ hyperparameter of Adam): 1e-8, 1e-4, 1e-2
- Shampoo:
  - – lr: 1e-5, 3e-5, 1e-4, 3e-4, 0.001, 0.003
  - – damping (i.e. $\epsilon$ in Algorithm 4): 1e-4, 3e-4, 0.001, 0.003, 0.01
- TNT:
  - – lr: 1e-7, 3e-7, 1e-6, 3e-6, 1e-5, 3e-5, 1e-4, 3e-4, 0.001
  - – damping (i.e. $\epsilon$ in Algorithm 3): 0.001, 0.003, 0.01, 0.03, 0.1, 0.3
- KFAC:
  - – lr: 1e-4, 3e-4, 0.001, 0.003, 0.01, 0.03, 0.1, 0.3
  - – damping: 0.01, 0.03, 0.1, 0.3, 1, 3, 10

The best hyper-parameter values determined by our grid searches are listed in Table 2.

### D.3 Experiment Settings for the CNN Problems

Table 3: Hyper-parameters (**initial** learning rate, weight decay factor) used to produce Figure 3 and the average validation accuracy across 5 runs with different random seeds shown in Figure 3

| Name | CIFAR-10 + ResNet32 | CIFAR-100 + VGG16 |
|---|---|---|
| TNT | (1e-4, 10) → 93.08% | (3e-5, 10) → 73.33% |
| KFAC | (0.01, 0.1) → 92.85% | (3e-4, 0.1) → 74.33% |
| Shampoo | (0.01, 0.1) → 92.63% | (0.003, 0.1) → 72.82% |
| Adam | (0.003, 0.1) → 92.92% | (3e-5, 10) → 72.27% |
| SGD-m | (0.03, 0.01) → 93.06% | (0.03, 0.01) → 73.44% |

Both CIFAR-10 and CIFAR-100 have 50,000 training data and 10,000 testing data (used as the validation set in our experiments). For all algorithms, we used a batch size of 128 at every iteration. In training, we applied data augmentation as described in [25], including random horizontal flip and random crop.

The ResNet32 model refers to the one in Table 6 of [19], whereas the VGG16 model refers to model D of [42], with the modification that batch normalization layers were added after all of the convolutional layers in the model.

It is worth noting that, in TNT and Shampoo, for the weight tensor in the convolutional layers, instead of viewing it as a 4-way tensor, we view it as a 3-way tensor, where the size of its 3 ways (dimensions) corresponds to the size of the filter, the number of input channel, and the number of the output channel, respectively. As a result, the preconditioning matrices of TNT and Shampoo will come from the Kronecker product of three matrices, rather than four matrices.

Weight decay, which is related to, but not the same as $L_2$ regularization added to the loss function, has been shown to help improve generalization performance across different optimizers [29, 45]. In our experiments, we adopted weight decay for all algorithms. The use of weight decay for TNT and Shampoo is described in Algorithm 3 and Algorithm 4, respectively, and is similarly applied to KFAC. Also note that weight decay is equivalent to $L_2$ regularization for pure SGD (without momentum). However, the equivalence does not hold for SGD with momentum. For the sake of a fair comparison, we also applied weight decay for SGD-m.

For TNT and Shampoo, we set $\epsilon = 0.01$. We also tried values around 0.01 and the results were not sensitive to the value of $\epsilon$; hence, $\epsilon$ can be set to 0.01 as a default value. For KFAC, we set the overall damping value to be 0.03, as suggested in the implementation in https://github.com/alecwangcq/KFAC-Pytorch. We also tried values around 0.03 for KFAC and confirmed that 0.03 is a good default value.

In order to obtain Figure 3, we first conducted a grid search on the initial learning rate (lr) and weight decay (wd) factor based on the criteria of maximizing the classification accuracy on the validation set. The range of the grid searches for the algorithms in our tests were:

- SGD-m:

  - lr: 3e-5, 1e-4, 3e-4, 0.001, 0.003, 0.01, 0.03, 0.1, 0.3, 1

  - wd: 0.001, 0.01, 0.1, 1

- Adam:

  - lr: 1e-6, 3e-6, 1e-5, 3e-5, 1e-4, 3e-4, 0.001, 0.003, 0.01, 0.03

  - wd: 1e-4, 0.001, 0.01, 0.1, 1, 10, 100

- Shampoo:

  - lr: 3e-5, 1e-4, 3e-4, 0.001, 0.003, 0.01, 0.03, 0.1

  - wd: 0.01, 0.1, 1, 10

- TNT:

  - lr: 1e-6, 3e-6, 1e-5, 3e-5, 1e-4, 3e-4, 0.001

  - wd: 1, 10, 100

- KFAC:

  - lr: 3e-6, 1e-5, 3e-5, 1e-4, 3e-4, 0.001, 0.003, 0.01, 0.03

  - wd: 0.01, 0.1, 1

The best hyper-parameter values, and the validation classification accuracy obtained using them, are listed in Table 3.

## D.4    A Comparison between TNT and TNT-EF

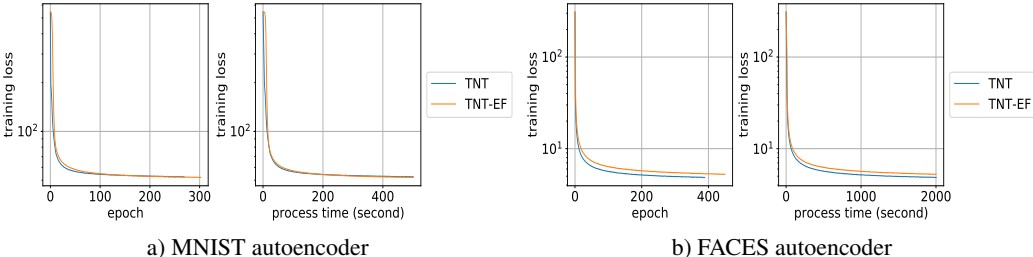

a) MNIST autoencoder                    b) FACES autoencoder

Figure 4:    Optimization performance comparison of the TNT and TNT-EF algorithms on two autoencoder problems.

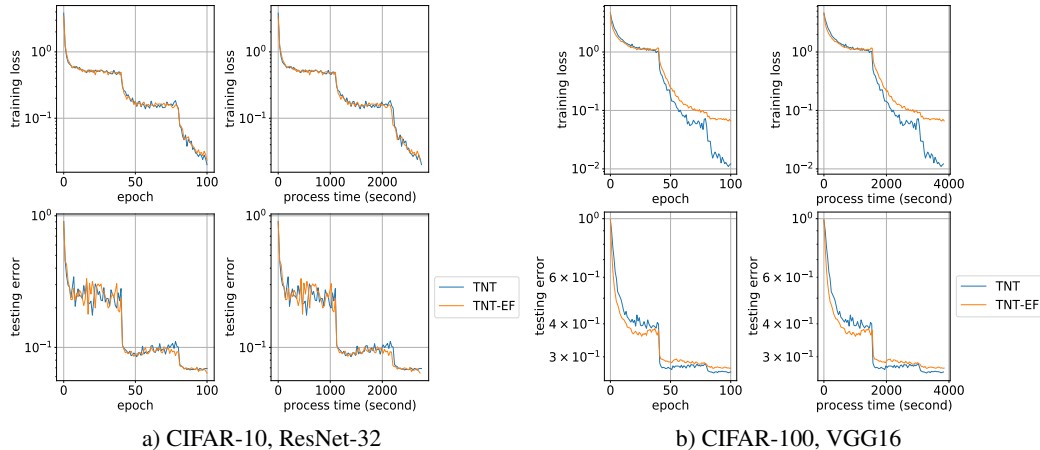

a) CIFAR-10, ResNet-32        b) CIFAR-100, VGG16

Figure 5: Generalization ability comparison of the TNT and TNT-EF algorithms on two CNN models. The upper row depicts the training loss, whereas the lower row depicts the validation classification error.

Table 4: Hyper-parameters (learning rate, damping) used to produce Figure 4

| Name | MNIST | FACES |
|---|---|---|
| TNT-EF | (3e-6, 0.01) | (3e-6, 0.01) |

Table 5: Hyper-parameters (**initial** learning rate, weight decay factor) used to produce Figure 5

| Name | CIFAR-10 + ResNet32 | CIFAR-100 + VGG16 |
|---|---|---|
| TNT-EF | (1e-4, 10) → 93.62% | (3e-6, 100) → 72.85% |

In this subsection, we compare our proposed TNT algorithm against a variant of it, TNT-EF, which uses an empirical Fisher (EF) preconditioning matrix in place of the true Fisher matrix. In other words, TNT-EF does everything specified in Algorithm 3, except that it does not perform the extra backward pass in Line 7 of Algorithm 3. When updating the matrices $\widehat{G_l^{(i)}}$, TNT-EF uses the empirical minibatch gradient, rather than the sampling-based minibatch gradient, i.e. the one coming from the extra backward pass.

We conducted a hyper-parameter grid search for TNT-EF, following the same procedure as the one that was used for TNT, whose performance was plotted in Figures 2 and 3. The best values for the TNT-EF hyper-parameters that we obtained are listed in Tables 4 and 5. We then plotted in Figures 4 and 5, the performance of TNT-EF, along with that of TNT, using for it the hyper-parameters given in Tables 2 and 3. As shown in Figures 4 and 5, TNT performed at least as well as TNT-EF, on the MNIST and CIFAR-10 problems, and performed somewhat better on the FACES and CIFAR-100 problems, which confirms the widely held opinion that the Fisher matrix usually carries more valuable curvature information than the empirical Fisher metric.

### D.5 More on Hyper-parameter Tuning

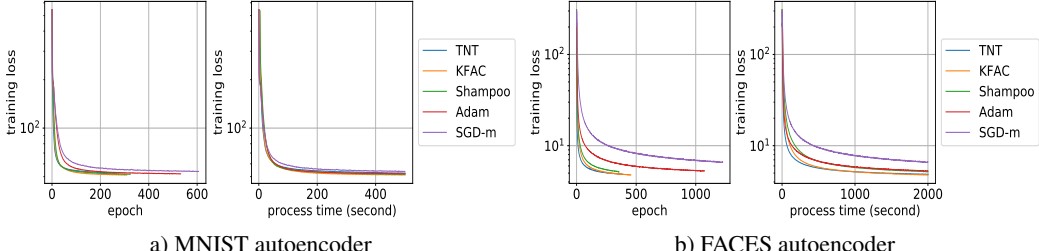

a) MNIST autoencoder                    b) FACES autoencoder

Figure 6: Optimization performance of TNT, KFAC, Shampoo, Adam, and SGD-m on two autoencoder problems, with more extensive tuning

Table 6: Hyper-parameter values used to produce Figure 6

| Problem | Algorithm | (learning rate, damping, $\mu$, $\beta$) |
|---------|-----------|------------------------------------------|
| MNIST | TNT | (1e-4, 0.1, 0.9, 0.9) |
| MNIST | KFAC | (3e-5, 0.01, 0.999, 0.999) |
| MNIST | Shampoo | (1e-4, 3e-4, 0.99, 0.99) |
| MNIST | Adam | (1e-4, 1e-4, 0.99, 0.99) |
| MNIST | SGD-m | (0.001, -, 0.99, -) |
| FACES | TNT | (1e-6, 0.003, 0.9, 0.9) |
| FACES | KFAC | (0.01, 3, 0.99, 0.99) |
| FACES | Shampoo | (1e-4, 3e-4, 0.99, 0.999) |
| FACES | Adam | (1e-4, 1e-4, 0.9, 0.9) |
| FACES | SGD-m | (0.001, -, 0.9, -) |

In this subsection, we expand on the experiments whose results are plotted in Figure 2, by incorporating the tuning of more hyper-parameters. To be more specific, we tuned the following hyper-parameters jointly:

1. SGD-m: learning rate and $\mu$;

2. all other algorithms[8]: learning rate, damping, $\mu$, and $\beta$.

The searching range for learning rate and damping is the same as in Sec D.2, whereas the searching range for $\mu$ and $\beta$ were set to be $\{0.9, 0.99, 0.999\}$. The obtained values for the hyper-parameters are listed in Table 6.

Figure 6 depicts the performance of different algorithms with hyper-parameters obtained from the aforementioned more extensive tuning process. Comparing the performance of different algorithms in Figure 6, we can see that the observations we made from Figure 2 still hold to a large extent. Moreover, with extensive tuning, second-order methods seem to perform similarly with each other, and are usually better than well-tuned first order methods on these problems.

As a final point, we would like to mention that one could also replace the constant learning rate for all of the algorithms tested with a "warm-up, then decay" schedule, which has been shown to result in good performance on these problems in [3]. Also, one could perform a more extensive tuning for the CNN problems. In particular, one could tune the initial learning rate, weight decay factor, damping, $\mu$, and $\beta$ jointly for the CNN problems.

See more in [8, 40] for the importance and suggestions on hyper-parameter tuning. Moreover, see [2] for other relevant numerical results, in particular for KFAC and Shampoo. In [2], KFAC is shown to work extremely well with a higher frequency of inversion, another direction for experiments that could be explored.

---

[8]For Adam, $\mu$ and $\beta$ refer to $\beta_1$ and $\beta_2$, respectively.