# OpenReview forum: "Tensor Normal Training for Deep Learning Models"
_NeurIPS.cc/2021/Conference — NeurIPS 2021 Spotlight_

### Official Review · Reviewer_5D53 · 2021-07-07

**Rating:** 7
**Confidence:** 5

**Summary:**

This paper presents TNT, a variant of Shampoo, but based on natural gradients. The key difference is that while Shampoo uses the gradient itself to compute the preconditioning matrices, TNT uses the gradients based on the predicted distribution to compute preconditioners - this is similar to KFAC. The paper compares TNT with Shampoo and KFAC, and the main differences are:
1. Shampoo preconditioners correspond to Empirical Fisher, whereas TNT approximates the true Fisher, at the cost of an extra backprop step. There is some evidence that the true Fisher captures curvature information better.
2. Shampoo preconditions the gradient as G x_1 U1^{-1/2k} x_2 U2^{-1/2k} ..., whereas TNT preconditions it as G x_1 U1^{-1} x_2 U2^{-1} ...  (modulo some damping).  Thus TNT needs only inverses of matrices, which is an easier operation than finding matrix roots.
3. KFAC approximates each layer's Fisher matrix as a Kronecker product using a unique algorithm, thus KFAC needs to be re-derived for each layer type. TNT like Shampoo works for all layer types.

The paper includes a theoretical analysis of convergence of TNT, as well as some experiments showing that it is competitive on some small datasets.

**Main Review:**

The algorithm should be of considerable interest to the ML community, as it seems to combine desirable attributes of both KFAC and Shampoo.

In section 3.3, it seems that the only reason one has to worry about the covariance parameters is the presence of damping. If damping is small (i.e. can be ignored), then can one can simplify the expression G x_1 U1^{-1} x_2 U2^{-1} ... to G x_1 G^(1)^{-1} x_2 G^(2)^{-1} ... * Tr(G^(1))^{k-1} ?

The experimental section could use a lot of improvements. It would be preferable to get a stronger CIFAR-10 baseline - SGD-m can reach much better accuracy than the 93.06 considered in this paper. The other datasets are also rather small, and it seems that the Shampoo authors have already addressed the performance issues in [3] --- for some reason this paper used SVD to compute matrix roots (including a data transfer to the CPU), whereas coupled Newton is much faster and can be executed on GPUs, so KFAC and Shampoo look artificially worse than TNT. Also this paper's results on auto-encoders are not nearly as good as [3] for these other optimizers.

Minor nit - the citations in the Appendix do not match with the main paper.

Update after author comments: The authors have addressed many of the comments about the experimental section, and will update this section in the paper. Correspondingly the overall assessment has been updated.

**Time Spent Reviewing:**

4

---

> ### Author Response · Authors · 2021-08-10
> **Thank you for your review!**
>
> We address here the specific comments, observations, and suggestions relevant to Reviewer 5D53’s review.
>
> #### 1. Expression when damping is ignored:
>
> The reviewer’s observation is correct. However, we note that damping appears to be used in all practical second-order methods. Hence, we believe that there is merit in proposing a representation of the tensor normal model relevant to optimization that is different from the other existing representations described in Section 3.3.
>
>
> #### 2. “stronger CIFAR-10 baseline”:
>
> We agree with the reviewer that SGD-m can reach better accuracy than 93.06%. In this paper, we fixed our model to be the original ResNet32 model (He et al 2015), which achieved 7.51% error (i.e. 92.49% accuray). This has also been validated by others, for example, in a well recognized github repo (linked omitted because of policy), the error is 7.37% (i.e. 92.63% accuracy).
>
> On the other hand, better accuracy is usually achieved with more advanced models developed on top of the original ResNet model. Even with ResNet32, better accuracy is usually achieved by using wider layers (e.g. Zhang et al 2019). However, we still appreciate the reviewer for bringing this up and if there is time, we will add experiments on using CIFAR-10 with other more recent models.
>
>
>
> #### 3. Coupled Newton:
>
> Please see the overall Official Comment.
>
>
> #### 4. Other concerns on the implementation of Shampoo and KFAC:
>
> #### 4.1. transferring data to CPU when doing SVD in Shampoo
>
> We thank the reviewer for carefulling looking into our code and raising the issue of transferring data to the CPU to perform SVD. The reason we did this in our code is as follows:
>
> There is a known issue with PyTorch that when SVD is performed on a GPU, sometimes the operations fail unexpectedly even if the matrix is positive definite (link to github issue omitted because of policy). In fact, there is a well recognized independent PyTorch implementation of Shampoo in github (link omitted because of policy ), which specifically transfers the data to the CPU to perform SVD whenever SVD is conducted.
>
> In our code, we ONLY transfer the data to the CPU to compute SVD when the SVD operation on the GPU fails, because: 1) we do not want to directly terminate the whole training process of Shampoo and call it a failure since this would be unfair to Shampoo; 2) we do not want to ALWAYS perform SVD on CPU, as this would definitely be slower and also unfair to Shampoo.
>
> Additionally, the failure of SVD on the GPU actually happened very rarely. In fact, it NEVER happened with Shampoo when the tuned hyper-parameter values were used. As a result, this does not lead to any disadvantage in the results presented in the paper. (It only happens (rarely) when hyper-parameters are set poorly.)
>
> Lastly, the issue of transferring data to the CPU is no longer a problem in our NEW runs of  Shampoo with coupled Newton, which will be included in our revision.
>
> #### 4.2. using SVD in KFAC:
>
> Note that using SVD instead of the matrix inverse in KFAC could yield different preconditioning matrices. Because if one uses matrix inverse, the overall damping term has to be split between the two KFAC matrices, which is an additional level of approximation, whereas if SVD is used, one can directly add the overall damping term to the Kronecker product (with the help of SVD).
> As discussed in Section D.1.2, we followed the implementation of KFAC in Zhang et al 2019 (see the github link in Section D.1.2), which uses SVD instead of matrix inverse. In fact, we tested both SVD and matrix inverse versions of KFAC in our experiments and found that only SVD could achieve the level of accuracy reported in our paper. Hence, we chose to use the SVD version in our paper, although it is indeed slower than matrix inverse.
>
> In fact, we spent a lot of effort choosing between different versions of KFAC to make sure we reported results from the best performing variant among them.
>
> Lastly, we do not think it is a good idea to use coupled Newton in KFAC for our experiments, because as in using the matrix inverse, one has to first split the damping term, which will not yield the same level of accuracy as SVD, as we stated above.
>
>
> #### 5. “results on autoencoders are not nearly as good as [3] for these other optimizers”:
>
> Below we present training loss results from our paper and from Anil et al 2020. Note that Anil et al 2020’s results were obtained from the plots in their paper.
>
> |               | Shampoo with SVD | Shampoo with coupled Newton | TNT        | KFAC | Shampoo (Anil 2020) | KFAC (Anil 2020) |
> | ----------- | -----------             | -----------                    | ----------- | ----------- | ----------- | ----------- |
> | MNIST   | 56.6                 | 52.8                          | 52.5       | 53.9       | 50 ~ 60       | 50 ~ 60       |
> | FACES  | 9.9                             | 6.58                                       | 4.8        | 5.3     | ~6   | ~5        |
>
> The training loss achieved by the coupled Newton version of Shampoo improves significantly, mainly due to the fact that Shampoo is able to complete more epochs within the same total time. Another reason for the small gap between the coupled Newton version of Shampoo and the version in Anil et al 2020 may be due to the fact that we did not tune the overall power between 0.5 and 1, and we did not use the warmup scheme used in Anil er al 2020.
>
> For KFAC, we believe that our results are pretty similar to those in Anil et al 2020. We view the gap as negligible and suspect that it is due to some differences in implementation.
>
>
> #### 6. Citation number issue:
>
> We thank the reviewer for raising this issue. We will be sure to fix it in our paper. In this Official Comment, we refer to papers by author+year notation and provide a full reference below.
>
>
> #### References
>
> Anil et al 2020: R. Anil, V. Gupta, T. Koren, K. Regan, and Y. Singer. Scalable second order optimization for deep learning
>
> He et al 2015: K. He, X. Zhang, S. Ren, and J. Sun. Deep residual learning for image recognition
>
> Zhang et al 2019: G. Zhang, C. Wang, B. Xu, and R. Grosse. Three mechanisms of weight decay regularization

---

### Official Review · Reviewer_KJXA · 2021-07-15

**Rating:** 6
**Confidence:** 5

**Summary:**

Paper proposes a tractable natural gradient approximation. It uses the shampoo approximation algebra combined with exponents of -`1 as its approximating the Fisher. Empirical experiments were carried out in two tasks: 1) autoencoder as well 2) convolutional nets.

**Limitations And Societal Impact:**

Yes.

**Main Review:**

The approach is clever,  and is also a straightforward extension of Shampoo to use sampled gradients from model's predictive distribution for computing statistics for the preconditioner. Another difference is that of exponents - as Shampoo (Gupta et al) approximates full matrix adagrad and requires the exponents sum to -1/2 for the loewner order to hold, however, as authors note its tuned over in Anil et al 2020.

The main concern with the work (and reason for score) is in the experiment section on tuning, and implementation of Shampoo. The author's use SVD to compute inverse pth root which is known to be expensive, and Anil et al 2020 already have shown a simple iterate method (coupled newton iterations) that runs the inverse-pth root much faster (~4x) - quite as fast as inverse operation on GPUs/TPUs.  Moreover, switching to eigh will make the implementation faster than SVD as there is no need to compute U, and V^T separately, as the matrices are symmetric, and postive (semi) definite. Thus main concern, is that experimental section results are unfair to the baseline being compared.  Code for inverse pth root ( https://github.com/google-research/google-research/blob/master/scalable_shampoo/jax/shampoo.py#L340 )

Another question, is comparison against https://arxiv.org/pdf/2006.08877.pdf which is considered to be already faster than K-FAC is missing from the experimental section, which could strengthen the paper.

Minor point (as its a recent work): But mentioning here for correctness (https://arxiv.org/abs/2106.06199) has shown K-FAC to work extremely well on these problems when tuned really well; They find the algorithms are sensitive to how often inverse is run, is another confounding factor in empirical comparisons.


== update ==

I am leaning towards acceptance as authors promised to tune hyper-parameters including warm-up and decay for the baselines.


**Time Spent Reviewing:**

24

---

> ### Author Response · Authors · 2021-08-10
> **Thank you for your review!**
>
> We address here the specific comments, observations, and suggestions relevant to Reviewer KJXA’s review.
>
> #### 1. Coupled Newton:
>
> We thank the reviewer for pointing us to the exact implementation of coupled Newton in Shampoo. For detailed discussion on this issue, please see the overall Official Comment.
>
>
> #### 2. K-BFGS:
>
> The reasons that we did not include K-BFGS in our comparison are as follows:
>
> #### 2.1.
>
> In this paper, we focus on optimizers that use Fisher or empirical Fisher (EF) as the preconditioning matrix. K-BFGS uses an approximation to the Hessian as the preconditioning matrix, which is related to Fisher and EF, but less relevant.
>
> #### 2.2.
>
> Similar to KFAC, K-BFGS needs to be re-derived for each type of layer. In Goldfarb et al 2020, the algorithm is only proposed for dense layers. We note that in Ren and Goldfarb 2021, K-BFGS has been extended to conv layers. However, this is still not a complete algorithm; for example, it is not clear to us how to handle the parameters of a batch normalization layer within the K-BFGS framework.
>
> #### 2.3.
>
> In Anil et al 2020, the following statement is made: “We compared Shampoo with KFAC and K-BFGS for standard autoencoder tasks on MNIST, FACES and CURVES, and found that all second order algorithms performed approximately the same, and far better than first order optimizers.” Hence, we believe that Shampoo and KFAC are representative of the state-of-the-art second-order methods, and hence, we did not feel that adding a comparison to K-BFGS was necessary. Nevertheless, we will try to implement K-BFGS for the problems that we tested and include the results in the paper.
>
>
> #### 3. LocoProp:
>
> We thank the reviewer for bringing this paper (of which we were not previously aware) to our attention. We will add a comment to our revision, indicating that in Amid et al 2021, KFAC is shown to work extremely well with a higher frequency of inversion.
>
>
> #### 4. Tuning
>
> Lastly, for the autoencoder problems, as noted in Anil et al 2020, setting the overall power of the preconditioner to 1 (instead of 1/2) could potentially improve the performance of Shampoo. In designing our experiments, we viewed this as requiring the tuning of an additional hyper-parameter and chose not to include it. We will make this clear in our revision.
>
>
> #### References
>
> Amid et al 2021: Amid, E., Anil, R., & Warmuth, M. K. LocoProp: Enhancing BackProp via Local Loss Optimization
>
> Anil et al 2020: R. Anil, V. Gupta, T. Koren, K. Regan, and Y. Singer. Scalable second order optimization for deep learning
>
> Goldfarb et al 2020: D. Goldfarb, Y. Ren, and A. Bahamou. Practical quasi-newton methods for training deep neural networks
>
> Ren and Goldfarb 2021: Ren, Y., & Goldfarb, D. Kronecker-factored quasi-newton methods for convolutional neural networks

---

> > ### Comment · Reviewer_KJXA · 2021-08-25
> > **Tuning guidelines**
> >
> > Thanks for the update on the paper.
> >
> > 1. Thanks for fixing the implementation, as it has changed the results completely.  It looks like there is no clear winner with the latest experiments. I would like to check if the authors agree that original claim that "TNT exhibited superior optimization performance to KFAC and Shampoo" is not true?
> >
> > 2.1 & 2.2 I do not think authors use batch norm in these models, as otherwise similar argument can be made for exclusion of K-FAC as well, but authors do include it in the comparisons. Also, this leads to the following question; are batch norm excluded from the standard resnet32?
> >
> > 4. My concern with the current set of results is the lack of tuning of baselines. Would authors be able to tune with a fixed: learning rate warmup, and a decay, and sweep hyper-parameters of both first order and second order methods i.e. (beta1, beta2, epsilon) ? Epsilon is a highly important hyper-parameter, fixing it does not lead to fair comparisons.

---

> > > ### Author Response · Authors · 2021-08-30
> > > **Thank you for your additional feedback!**
> > >
> > > We thank the reviewer for providing additional feedback. We also thank the reviewer for confirming that we have addressed the concerns raised in the reviewer’s initial review. Below we address Comments 1., 2., and 3. of 25 Aug 2021.
> > >
> > >
> > > #### 1. Descriptions in the abstract
> > >
> > > We thank the reviewer for revisiting the descriptions in the abstract. We propose to change the sentence
> > >
> > > “In our experiments, TNT exhibited superior optimization performance to KFAC and Shampoo, and to state-of-the-art first-order methods.”
> > >
> > > to
> > >
> > > “In our experiments, TNT exhibited superior optimization performance to state-of-the-art first-order methods, and comparable optimization performance to the state-of-the-art second-order methods KFAC and Shampoo.”
> > >
> > > Note: we may have to slightly modify the above wording if new experiments requested by the reviewer lead to changes in the relative performance of the methods, which we think is unlikely.
> > >
> > >
> > > #### 2. K-BFGS, KFAC, and batch norm
> > >
> > > We thank the reviewer for diving deeply into the setting of the experiments. To clarify, we did use batch norm (BN) in the CNN experiments in our paper (i.e. a. and b. of Figure 3) for all algorithms that we compared, as we mentioned in Section D.3.
> > >
> > > Moreover, as for K-BFGS and KFAC, we agree with the reviewer that similar argument could be made for KFAC, as the original KFAC papers (Martens and Grosses 2015 and Grosse and Martens 2016) did not specify how to handle BN layers. However, as mentioned in Section D.1.2, “For the parameters in the BN layers, we used the gradient direction, exactly as in [github link omitted].” Note that using the gradient direction for BN layers in KFAC was mentioned in the Zhang et al 2019a (i.e. a follow-up paper about KFAC) and we simply followed their choice. However, to the best of our knowledge, we are not aware of any paper addressing how to deal with BN layers in K-BFGS and developing an approach to do so is beyond the focus of our paper.
> > >
> > > Nevertheless, as mentioned in our initial Official Comment, we will try to implement K-BFGS for the problems that we tested (probably using the gradient direction for BN layers as in KFAC).
> > >
> > > Finally, to answer the last question, we believe that BN is included in the standard ResNet32, as described in He et al 2015, and we followed this choice in our experiments.
> > >
> > >
> > > #### 3. Tuning
> > >
> > > We thank the reviewer for laying out detailed suggestions for tuning. Although the word “tuning” was mentioned in the initial review, there were no specific suggestions on what to tune. The only issue raised concerned using SVD versus coupled Newton, which we addressed by making the necessary changes to our code for Shampoo and described in our Official Comments.
> > >
> > > #### 3.1. Some clarifications
> > >
> > > First, we would like to make several clarifications w.r.t. tuning.
> > >
> > > #### 3.1.1. Learning rate decay
> > >
> > > For the CNN experiments in our paper, we conducted a multi-step learning rate decay schedule for all algorithms (see Section 6.2), as in Zhang et al 2019a, Zhang et al 2019b, etc. Admittedly, this is different from the reviewer’s proposal of implementing a warm up followed by a decay. However, we believe that the multi-step learning rate decay schedule that we used is a standard choice for CIFAR-10/Resnet32 and CIFAR-100/VGG16. For the autoencoder experiments, we use a fixed learning rate, as in Goldfarb et al 2020, George et al 2018, etc.
> > >
> > > #### 3.1.2. epsilon
> > >
> > > For the autoencoder experiments, we tuned the epsilon (see Section D.2), which, as the reviewer mentioned, is important for obtaining a fair comparison. For the CNN experiments, we used a fixed epsilon (or damping value) for all algorithms besides SGD-m, as in Zhang et al 2019a and Zhang et al 2019b. As mentioned in Section D.3, for TNT and Shampoo, “we also tried values around 0.01 and the results were not sensitive to the value of epsilon”, and similar observations were made for KFAC in Section D.3.
> > >
> > > #### 3.2. How we chose our tuning procedure in our initial submission
> > >
> > > For the autoencoder experiments in our paper, we largely followed the tuning procedure in Goldfarb et al 2020, which includes the simultaneous tuning of learning rate and epsilon for all applicable algorithms.
> > >
> > > For the CNN experiments in our paper, we largely followed the tuning procedure in Zhang et al 2019a and Zhang et al 2019b, which tuned the (initial) learning rate and weight decay factor, and set the other hyper-parameters (including epsilon/damping, beta1, and beta2) to default values.
> > >
> > > Admittedly, the reviewer’s proposal to tune beta1, beta2, epsilon, learning rate, and weight decay (for CNN problems only) is indeed more comprehensive than what we did. To the best of our knowledge, if one goes through related papers, a comparable level of such extensive tuning only appears in Anil et al 2020. Most other papers (including our paper) usually only tune a few key hyper-parameters. For instance, in Gupta et al 2018, alpha (beta2 in our paper) is set to a default value of 0.9, and there is no mention of whether and how epsilon is tuned.
> > >
> > >
> > >
> > >
> > >
> > > #### 3.3. Our plan
> > >
> > > In spite of our comments above, we appreciate the reviewer’s suggestion regarding more comprehensive tuning. If there is time and sufficient computing resources available, we will try to include more comprehensive tuning of baselines in our revision.
> > >
> > > From our understanding, the reviewer’s suggestions require tuning 4 hyper-parameters for the autoencoder experiments and 5 hyper-parameters for the CNN experiments. Since we submitted our initial Official Comments on Aug. 10, received the reviewer’s reply on Aug. 25, and the discussion period will end at the start of Sept., it is unlikely that we will be able to communicate to the reviewer our new results with more comprehensive tuning within the discussion period, mainly due to the limited time/computing/manpower resources.
> > >
> > > We lay out in the following our plan on tuning based on our understanding of the reviewer’s suggestions, and would appreciate it if the reviewer could confirm our understanding (see below).
> > >
> > > #### 3.3.1. (fixed) learning rate warmup and decay
> > >
> > > By looking at Anil et al 2020 and the github repository that the reviewer pointed out on “scalable Shampoo”, we assume that “learning rate warmup and decay” refers to dividing the whole training procedure into two stages, warmup and decay. (It is unclear to us what the warmup parameter stands for in Table 3 of Anil et al 2020.) Hence, our plan is, with a given total epochs, using the first 10% epochs as warm up and the remaining 90% as decay; i.e., we will not tune the percent of epochs used for the two regimes. The other hyper-parameter is the (base) learning rate, which we will tune as described below.
> > >
> > > #### 3.3.2. sweeping hyper-parameters
> > >
> > > In addition to the warmup/decay schedule described above, we will tune the hyper-parameters:
> > >
> > > - beta1, beta2, epsilon, and (base) learning rate in the autoencoder experiments;
> > >
> > > - beta1, beta2, epsilon, (base) learning rate, and weight decay factor in the CNN experiments.
> > >
> > > We will use a grid search for beta1 and beta2 using the values {0.9, 0.99, 0.999}. For CNN experiments, we will search epsilon from the same range that we used for the autoencoder problems (see Section D.2). The other ranges will be the same as the ones we used in our initial submission (see Section D).
> > >
> > >
> > > #### 3.4. Final remarks
> > >
> > > Looking at the autoencoder experiments in Anil et al 2020, we noticed that this paper has already conducted an extensive hyper-parameter search for Shampoo, which includes “momentum” (not sure if this is beta1 or beta2 or both), learning rate, (ridge) epsilon, and warmup. The power alpha was an additional hyper-parameter that was set to a value in [0, 1]. For the other algorithms, this paper states that “(we) used the hyperparameters they (Goldfarb et al 2020) found to be optimal for each of these algorithms for each dataset”. As indicated in Goldfarb et al 2020, only learning rate and epsilon were tuned (beta1 and beta2 were fixed, and there was no warmup/decay schedule). We assume that this is also the case for the results reported in Figure 2 of Anil et al 2020.
> > >
> > > Looking at Figure 2 in Anil et al 2020, we observed that the highly-tuned Shampoo algorithm performs approximately the same as KFAC (and K-BFGS) algorithms, which, according to Anil et al 2020 and Goldfarb et al 2020, only involves tuning the learning rate and epsilon. On the other hand, as the results of the updated experiments reported in our initial Official Comments show, TNT, KFAC, and Shampoo have comparable optimization performance. Hence, we doubt that conducting more extensive hyper-parameter tuning will significantly change our current conclusions.
> > >
> > >
> > > #### References
> > >
> > > Anil et al 2020: R. Anil, V. Gupta, T. Koren, K. Regan, and Y. Singer. Scalable second order optimization for deep learning
> > >
> > > George et al 2018: T. George, C. Laurent, X. Bouthillier, N. Ballas, P. Vincent. Fast Approximate Natural Gradient Descent in a Kronecker Factored Eigenbasis
> > >
> > > Goldfarb et al 2020: D. Goldfarb, Y. Ren, and A. Bahamou. Practical quasi-newton methods for training deep neural networks
> > >
> > > Grosse and Martens 2016: R. Grosse and J. Martens. A kronecker-factored approximate fisher matrix for convolution layers
> > >
> > > Gupta et al 2018: V. Gupta, T. Koren, and Y. Singer. Shampoo: Preconditioned stochastic tensor optimization
> > >
> > > He et al 2015: K. He, X. Zhang, S. Ren, and J. Sun. Deep residual learning for image recognition
> > >
> > > Martens and Grosses 2015: J. Martens and R. Grosse. Optimizing neural networks with kronecker-factored approximate curvature
> > >
> > > Zhang et al 2019a: G. Zhang, C. Wang, B. Xu, and R. Grosse. Three mechanisms of weight decay regularization
> > >
> > > Zhang et al 2019b: Michael R. Zhang, James Lucas, Jimmy Ba, and Geoffrey E. Hinton. Lookahead optimizer: k steps forward, 1 step back

---

> > > > ### Comment · Reviewer_KJXA · 2021-08-30
> > > > **Response on tuning**
> > > >
> > > > Here are works that describe tuning of baselines:
> > > >
> > > > [1] "On Empirical Comparisons of Optimizers for Deep Learning", Dami Choi, Christopher J. Shallue, Zachary Nado, Jaehoon Lee, Chris J. Maddison, George E. Dahl  ​https://arxiv.org/abs/1910.05446
> > > >
> > > > [2] More  recently tuning concerns and leading to incorrect conclusion have been discussed in:
> > > > Descending through a Crowded Valley — Benchmarking Deep Learning Optimizers, Robin M. Schmidt, Frank Schneider, Philipp Hennig
> > > >
> > > > It should be reasonable ask of any optimization work to tune the baselines well. This includes tuning of hyper-parameters as well, as picking a reasonable warmup/decay schedule.

---

> > > > > ### Author Response · Authors · 2021-08-31
> > > > > **Thank you for your reply!**
> > > > >
> > > > > We thank the reviewer for pointing us to papers discussing tuning of baselines. We will add these references to our revision and stress the importance of baseline tuning, a view that we share with the authors of these papers and the reviewer.
> > > > >
> > > > > As we mentioned in our previous reply, although it is unlikely (actually impossible) for us to finish such an extensive tuning within the discussion period, we will try our best to tune the hyper-parameters, as well as picking a reasonable warmup/decay schedule for each of the baselines, and include the results in our revision.

---

### Official Review · Reviewer_kZ6w · 2021-07-18

**Rating:** 7
**Confidence:** 4

**Summary:**

The method proposes a novel approximate method, for approximating the Fisher matrix of a neural network, which to be used as a preconditioner for second-order optimization during training. The method, termed Tensor Normal Training (TNT) approximates the diagonal blocks of the Fisher via a Tensor Normal distribution. The idea of treating second order methods in this fashion (as having a Gaussian approximation to the parameters) has other connections to Bayesian learning methods and interpretations of Natural Gradient, e.g. like [1]. The key part of the method is to use a Kronecker Factored gaussian distribution over each rank of the actual variable tensor (e.g. Matrix Normal for a matrix weights). The authors derive the exact computation needed for estimating the covariance matrices show the convergence properties of the method. They relate the method to other well known approximate curvature matrix methods - Shampoo and KFAC - and compare on two autoencoder tasks and two classification tasks, where the results suggest that TNT practically outperforms the other methods, with better training and comparable test performance.

[1] - Khan, M.E. and Rue, H., 2021. The Bayesian Learning Rule. arXiv preprint arXiv:2107.04562.


**Main Review:**

The paper is written really well, very clear simple idea which shows some promise. I liked the idea a lot and think it is very clever. The significance of the method is less clear, as although the authors show benefit on several problems, these are achieved under somewhat restricted conditions - small batch size, fixed learning rate and damping throughout optimization, in contrast to for instance the full fledged KFAC algorithm from [33]. Nevertheless, I understand including all that can be quite difficult to do, hence I see this as a minor point, but hope the authors to consider making this comparison in the near future. The actual method, I find very interesting, as for dense layers it is almost exactly the same as KFAC, but not quite. In fact, I would urge the authors to have a plot/figure, where for a single layer where they can compute the Full fisher exactly, what are the difference between the KFAC and TNT approximation (specifically of the eigen directions) e.g. more than the just the cosine similarity. I do not have a clear intuition on this, but I think it would be very useful for understanding why TNT might be a better approximation.

I did not fully check the maths in the appendix regarding the convergence rate theorems and lemmas, but they look correct at a glance.

A few other concerns on the experimental section:

1. The autoencoder examples seem to be with ReLU where previous works have always done this with hyperbolic tangent activation functions. I would urge the authors to rerun the experiments in this way, as one of the key points of these experiments is that they are very difficult from optimization point of view and the activation matters.

2. Could you also include an experiment where you use TNT for the joint weights+biases parameters, where you concatenate the biases to the weights to form a new matrix.

3. For the convolutions could you include a TNT where it treats the parameters the same way as KFAC - as a single matrix. This would server as making more clear to the reader whether the benefit of TNT comes from better factorization of the approximate matrix, or from better estimation of the factors themselves, or from both.

Nit pick: Note that with the appropriate normalization of the two factors in KFAC (e.g. rescaling them such that they have the same norms), the way it computes a separate damping for each factor, can be reduced indeed to being the same for both. Would be nice if you can include this in the appendix, to make the point more clear to a curious reader.



**Time Spent Reviewing:**

3

---

> ### Author Response · Authors · 2021-08-10
> **Thank you for your review!**
>
> We address here the specific comments, observations, and suggestions relevant to Reviewer kZ6w’s review.
>
> #### 1. Comments about small batch size, fixed learning rate:
>
> #### 1.1. Small batch size:
>
> For the autoencoder problems, we used a batch size of 1000 (as in Botev et al 2017, Goldfarb et al 2020, and Anil et al 2020), which is  a moderate size. For the CNN problems, we used a batch size of 128 (a standard setting as in Zhang et al 2019, etc), which is indeed a relatively  small batch size. It would be interesting to also explore larger batch sizes for these CNN problems.
>
> #### 1.2. Fixed learning rate:
>
> For the autoencoder problems, we used a fixed learning rate, as in Goldfarb et al 2020, George et al 2018, etc. For the CNN problems, we used a standard learning rate decay schedule (see the paragraph starting on line 286). Also note that the values in Table 3 (in the Appendix) are the initial learning rates, which is reduced by a factor of 0.1 every 40/80 epochs.
>
>
> #### 2. “The Bayesian Learning Rule.”:
>
> We thank the reviewer for bringing this to our attention. We will add a brief discussion to the paper on the connection between the natural gradient method and Bayesian learning.
>
>
> #### 3. “somewhat restricted conditions - damping throughout optimization”:
>
> We assume that the reviewer is referring to the approach for automatically adapting the damping term (i.e. the LM approach) in Martens and Grosse 2015. The reasons that  we decided to use a fixed damping are as follows:
>
> #### 3.1.
>
> We would like to focus on comparing different preconditioning methods. Including the LM approach (which could be applied to KFAC, TNT and Shampoo) is likely to add more confounding factors and hide the actual comparison between preconditioners.
>
> #### 3.2.
>
> To the best of our knowledge, the most modern implementation of KFAC (such as the official TensorFlow KFAC code (link omitted because of policy) and the KFAC implementation in Zhang et al 2019) usually treats damping as a hyper-parameter, rather than modifying the value of the damping parameter during the training process.
>
> #### 4. ReLU in autoencoder:
>
> We chose to use ReLU because it is more widely used nowadays. Moreover, this activation function is also used in Goldfarb et al 2020 and Anil et al 2020. (Since Anil et al 2020 used the tuned hyper-parameter given in Goldfarb et al 2020, we assume that they did not change the activation function to the hyperbolic tangent function.) However, we agree that using the hyperbolic tangent function would make the problems harder and perhaps provide a more definitive comparison between the optimizers. We will add a set of experiments which use hyperbolic tangent activation to our paper.
>
>
> #### 5. Matrix rescaling in KFAC:
>
> We will include this discussion in the Appendix.
>
>
> #### 6. “joint weights+biases parameters”, “treats the parameters in TNT the same way as KFAC”, and “plot/figure, where for a single layer...”:
>
> These are really valuable suggestions. If there is time, we will consider exploring these variants and adding comparisons to the Appendix of our paper.
>
>
> #### References
>
> Anil et al 2020: R. Anil, V. Gupta, T. Koren, K. Regan, and Y. Singer. Scalable second order optimization for deep learning
>
> Botev et al 2017: A. Botev, H. Ritter, and D. Barber. Practical gauss-newton optimisation for deep learning
>
> George et al 2018: T. George, C. Laurent, X. Bouthillier, N. Ballas, and P. Vincent. Fast approximate natural gradient descent in a kronecker factored eigenbasis
>
> Goldfarb et al 2020: D. Goldfarb, Y. Ren, and A. Bahamou. Practical quasi-newton methods for training deep neural networks
>
> Martens and Grosse 2015: J. Martens and R. Grosse. Optimizing neural networks with kronecker-factored approximate curvature
>
> Zhang et al 2019: G. Zhang, C. Wang, B. Xu, and R. Grosse. Three mechanisms of weight decay regularization

---

### Author Response · Authors · 2021-08-10
**Overall Official Comment**

We would like to thank all the reviewers for their insightful reviews and helpful comments, and in particular, their suggestions for improving the experimental section of our paper. In this overall official comment, we focus on addressing the implementation of Shampoo and the issue of using the coupled Newton method, as these were perhaps the principal points raised by two of the reviewers. We refer all of the reviewers to the official comments to their reviews for our response to other issues that they raised.

We have rerun the experiments of Shampoo with coupled Newton, and we **will** update the results in our paper (as well as stress the improvement achieved by using coupled Newton). We now describe the updated results.

#### 1. Updated results:

We treated Shampoo with coupled Newton as an independent optimizer and tuned it as we did for the other optimizers, as described in the paper. Since we cannot include figures in the official comments, we will try to describe our updated results in enough detail for it to be clear what the plots will look like in a revised version of our paper. Specifically, see the tables below.

#### 1.1. Using coupled Newton did indeed improve the speed of Shampoo.

#### 1.1.1. Autoencoder problems:

In these problems, as in our paper, we fixed the total process time that each algorithm was allowed. In the table below, we list the number of epochs that each of the three algorithms, Shampoo with SVD, Shampoo with coupled Newton, and TNT, completed.

|               | Shampoo with SVD | Shampoo with coupled Newton | TNT |
| ----------- | -----------                   | -----------                                      | ----------- |
| MNIST   | 103                          | 308                                             | 268       |
| FACES  | 89                             | 328                                            | 389        |

It is clear that Shampoo was at least 3 times faster when implemented with coupled Newton compared with SVD.  Also we can see from the table that TNT was faster (i.e., completed more epochs) than Shampoo with coupled Newton for FACES, but was slower for MNIST.

#### 1.1.2. CNN problems:

In these problems, as in our paper, we fixed the total number of epochs to be completed by each algorithm. In the table below, we list the total time (seconds) that each of the three algorithms used.

|                     | Shampoo with SVD | Shampoo with coupled Newton | TNT |
| -----------       | -----------                   | -----------                                     | ----------- |
| CIFAR 10    | 2744                         | 2692                                          | 2843       |
| CIFAR 100  | 8554                         | 4277                                           | 3840        |

On CIFAR 100 (with VGG16), Shampoo speeds up significantly with coupled Newton. There was not much difference between the process times of the coupled Newton and SVD implementations in Shampoo (and TNT as well) on CIFAR 10 (with ResNet32). The main reason for this is that this model is relatively small, so the inverse and inverse root operations did not take up a significant amount of time. Also, these operations were performed only every 100 iterations.

#### 1.2. Overall progress

#### 1.2.1. Autoencoder problems:

The following table lists the final training loss of the three optimizers on MNIST and FACES. In addition, the epoch-wise progress for the two versions of Shampoo were basically the same, but as mentioned above, coupled Newton completes more epochs.

|               | Shampoo with SVD | Shampoo with coupled Newton | TNT |
| ----------- | -----------                   | -----------                                      | ----------- |
| MNIST   | 56.6                         | 52.8                                             | 52.5       |
| FACES  | 9.9                           | 6.58                                             | 4.8        |

#### 1.2.2. CNN problems:

The following table lists the accuracy achieved by the three optimizers on CIFAR 10 and CIFAR 100. Since we fixed the number of training epochs to be 100, we observed that the two versions of Shampoo had similar epoch-wise progress and that the final accuracy achieved was very similar.

|                     | Shampoo with SVD | Shampoo with coupled Newton | TNT |
| -----------       | -----------                   | -----------                                     | ----------- |
| CIFAR 10    | 92.70%                    | 93.06%                                       | 93.08%       |
| CIFAR 100  | 73.09%                    | 72.77%                                        | 73.33%        |

#### 1.3.

Summarizing the above results, we see that using coupled Newton definitely speeded up Shampoo, while not significantly affecting the accuracy achieved on the CNN problems. Also, our new results show that TNT outperforms Shampoo with coupled Newton (albeit only slightly on MNIST, CIFAR 10, and CIFAR 100, and to a somewhat greater degree on FACES).




#### 2. Why we did not test Shampoo with coupled Newton in our submitted paper:

#### 2.1.

Our experiments were designed to compare different preconditioning methods, while introducing as few confounding factors as possible. Since coupled Newton can be applied to computing both the inverse root and inverse of a matrix, and hence could be used with both KFAC and TNT (which we do not plan to implement), we chose to implement the SVD version of Shampoo.

#### 2.2.

There are two additional hyper-parameters associated with coupled Newton, i.e. max iteration number and error tolerance. Our new results indicate that setting these to default values is good enough. This will be discussed in our revised paper.

#### 2.3.

We implemented all methods with PyTorch. SVD and matrix inverse are well supported in PyTorch, but not coupled Newton. (To obtain our new results, we used the code referred to by reviewer KJXA and made it a PyTorch code.)

#### 3.

Finally, we would like to emphasize that the purpose of our experiments was and is not to show that TNT is the best optimizer under all circumstances. Rather, our focus was and is on presenting a fair comparison between our new TNT method and current state-of-the-art optimizers applied to several commonly used problems in deep learning.

---

### Decision · Program_Chairs · 2021-09-27

**Decision:**

Accept (Spotlight)

**Comment:**

This paper develops a new 2nd-order optimization method targeted at neural networks that combines aspects of K-FAC and Shampoo. In particular, it uses a Shampoo-style approximation to estimate the Fisher information matrix instead of the Empirical Fisher (as in Shampoo), and like K-FAC raises the curvature matrix approximation to the power -1 to compute the preconditioner. The reviewers agree that this is a well written paper that makes a solid contribution to the area.

The main concerns of the reviewers are related to the experiments and the tuning of different optimizers. I share these concerns, but am pleased to see that the authors are quite willing to rerun their experiments based on the reviewer's guidance and update their conclusions accordingly. One thing I would like to bring up again is the use of SVD (which is much slower than inverses) and constant damping in K-FAC. Especially for those autoencoder problems a constant damping is highly suboptimal. The Tensorflow release of K-FAC has a MNIST autoencoder experiment implementation (under tensorflow_kfac/examples/autoencoder_mnist.py) which reproduces the results from the original paper (which gets a loss of 50.25 after 1.5k steps using an increasing batch size schedule). It would be quite interesting to see how well the proposed preconditioner would performed as a drop-in replacement for K-FAC in that setting, or even one with a fixed batch size of 60k.

Another thing probably worth pointing out is that this method, like Shampoo, uses the tensor shape of the network's parameters to construct its approximations. That this works well for convnets has entirely to do with the specific role of these parameters play in the network, and how this just happens to be related to their shape in typical implementations. In general, shape can be totally arbitrary, and there is nothing preventing me from parameterizing all of the convolutional layers in my network with flat vectors, or with tensors of the same shape but with randomly permuted entries.